# Can Language Models Laugh at YouTube Short-form Videos?

**Dayoon Ko**[1]    **Sangho Lee**[2]    **Gunhee Kim**[1]

[1]Seoul National University    [2]Allen Institute for Artificial Intelligence

dayoon.ko@vision.snu.ac.kr    sanghol@allenai.org    gunhee.kim@snu.ac.kr

https://github.com/dayoon-ko/ExFunTube

## Abstract

As short-form funny videos on social networks are gaining popularity, it becomes demanding for AI models to understand them for better communication with humans. Unfortunately, previous video humor datasets target specific domains such as speeches or sitcoms, and mostly focus on verbal cues. We curate a user-generated dataset of 10K multimodal funny videos from YouTube, called **ExFunTube**. Using a video filtering pipeline with GPT-3.5, we verify both verbal and visual elements contributing to humor. After filtering, we annotate each video with timestamps and text explanations for funny moments. Our ExFunTube is unique over existing datasets in that our videos cover a wide range of domains with various types of humor that necessitate a multimodal understanding of the content. Also, we develop a zero-shot video-to-text prompting to maximize video humor understanding of large language models (LLMs). With three different evaluation methods using automatic scores, rationale quality experiments, and human evaluations, we show that our prompting significantly improves LLMs' ability for humor explanation.

## 1  Introduction

Today, a huge number of short-form funny videos are popularly circulated on social media platforms. Although humor often triggers instant laughter, understanding humor is not a straightforward process. Numerous studies (Hazlitt, 1845; Kant, 1786; Nerhardt, 1970; Jones, 1970; Shultz, 1972; Suls, 1972, 1983) have explored the cognitive process of humor appreciation. For instance, Hazlitt (1845) and Kant (1786) propose the incongruity theory, asserting that incongruity provokes laughter. Nerhardt (1970) further develops the idea by defining the discrepancy between expectation and content, such as punchlines or cartoons. Suls (1972) suggests the incongruity-resolution theory, positing that humor arises only when the incongruity is resolved by

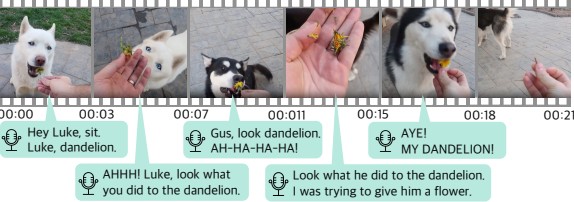

Timestamps & Explanations of the funny moments

| | |
|---|---|
| 2s ~ 4s | It's funny because the white dog is shown a dandelion and then the dog eats the dandelion unexpectedly. |
| 8s ~ 10s | It's funny because the black and white dog is shown a dandelion and does the same thing as the dog and eats the dandelion. Also, the man's intention was simply to give the dog a flower, not for the dog to eat it. |
| 17s ~ 20s | The dog turns and notices the dandelion, then goes over and eats the dandelion from the man's hand. It's funny because of the man's exaggerated reaction. |

Figure 1: An example from the ExFunTube dataset. We curate funny short-form videos in various domains through a filtering pipeline that verifies both verbal and visual elements contributing to humor. Each video is annotated with timestamps and explanations for funny moments. In this example, three funny moments are identified.

retrieving information from the joke, cartoon, or the perceiver's own knowledge. Since a sufficient understanding of the context is required to perceive and further resolve the incongruity, understanding humor can be challenging. Nevertheless, if AI models can understand humor, they could interact more effectively with humans by providing empathetic responses based on users' sense of humor. Furthermore, if the models understand short-form funny videos, they can recommend videos based on users' preferences or even generate witty titles based on video contexts.

Several studies (Hasan et al., 2019; Castro et al., 2019; Patro et al., 2021; Kumar et al., 2022) have collected humorous video datasets to investigate whether models can understand if a video is funny or not. However, the datasets have been gathered from a limited domain, such as speeches or sitcoms. For example, Hasan et al. (2019) collect videos from TED, where there is a single speaker, and visual cues are restricted to gestures or facial ex-

| Dataset | Modality | Type | #Data Points | Data Config | Exp | Task |
|---|---|---|---|---|---|---|
| ExPUN | T | Pun | 2K | {Pun, Keywords, Up to 5 scores & explanations} | ✓ | Pun *Exp* |
| AVH / FOR | I | Abstract Scene | 3K / 15K | {A funny image, An unfunny image, 10 funniness ratings} / {A counterpart (object replaced) image} | - | Image Humor Scoring & Altering |
| NYCC | I,T | Cartoon | 0.7K | {Cartoon, Three finalist captions, 3 annotations of locations, descriptions, uncanny descriptions, relevant entities, and explanations} | ✓ | Cartoon Caption *Exp* |
| MORE | I,T | Posts | 3K | {Image, Caption, 1 explanation} | ✓ | Image Sarcasm *Exp* |
| MUStARD | V,A,T | Sitcom | 6K | {Video, Binary (funny/unfunny) label} | - | Video Sarcasm *BC* |
| WITS | V,A,T | Sitcom | 2.2K | {Video, One Explanation} | ✓ | Dialogue Sarcasm *Exp* |
| UR-FUNNY | V,A,T | Speech | 8K | {Video, Binary (funny/unfunny) label} | - | Video Humor *BC* |
| MHD | V,T | Sitcom | 11K | {Video, Binary (funny/unfunny) label} | - | Video Humor *BC* |
| ExFunTube | V,A,T | Short-form Youtube videos | 10K | {Video, Up to 3 timestamps & explanations} | ✓ | Video Humor *Exp* |

Table 1: Comparison of our ExFunTube with previous humor datasets: ExPUN (Sun et al., 2022), AVH&FOR (Chandrasekaran et al., 2016), NYCC (Hessel et al., 2022), MORE (Desai et al., 2022), MUStARD (Castro et al., 2019), WITS (Kumar et al., 2022), UR-FUNNY (Hasan et al., 2019), and MHD (Patro et al., 2021) . In the Modality column, I, V, A, and T denote image, video, audio, and text, respectively. The #Data Points column shows only the number of positive (humorous) data points. The Data Config column specifies the composition of each data point. The Exp column indicates the presence of annotated explanations. In the Task column, *Exp* and *BC* are abbreviations of explanation generation and binary classification task each.

pressions. Castro et al. (2019) build the MUStARD dataset from four sitcoms, mainly from "Friends" and "Big Bang Theory," and Patro et al. (2021) collect the MHD dataset from the sitcom "Big Bang Theory." However, in sitcoms, the fixed actors follow a predetermined script on a constructed set, and the punchline plays a crucial role, so the visual elements may have less contribution to humor. Moreover, the aforementioned video datasets only have binary labels indicating whether the content is humorous or not. As binary classification may not evaluate whether a model truly understands the humor in a video, Kumar et al. (2022) collect WITS with annotated text explanations. However, this dataset is limited to sarcasm, a specific form of humor, and focuses on sarcasm explanation in dialogue. It highlights a need for a humor explanation dataset that considers visual elements more and covers general humor.

To this end, we curate **ExFunTube**, a dataset of funny, short-form videos with explanations. These videos are collected from user-generated YouTube videos, which are shared on the "r/youtubehaiku" subreddit. In this subreddit, users upload short-form funny videos, typically up to 30 seconds long. We develop a video filtering pipeline with GPT-3.5 (Ouyang et al., 2022), designed to exclude the videos with minimal visual impact on humor. Then, we annotate the collected videos with timestamps and text explanations of funny moments, as exemplified in Figure 1.

Recent LLMs show great performance for ex-

plaining humor present in text to some extent (Chowdhery et al., 2022). Inspired by the recent research on multimodal-informed prompting (Zeng et al., 2022), we convert video content into text, leveraging various zero-shot models on diverse modalities of the video. We provide LLMs with the text prompt as a linguistic summary of video content. Specifically, we consider two modalities of the video content: visual and audio. From the visual modality, we obtain dense video descriptions. From the audio modality, we acquire speech transcripts and sound labels. Finally, we chronologically integrate them into a text prompt that can maximize LLMs' ability for humor explanation.

Since evaluating a model's ability to explain humor is challenging, we report our results in three different ways: model-based automatic scores, rationale quality metrics with the moment localization task, and human evaluation. First, we report model-based metrics instead of those using word overlap. Second, we conduct a rationale quality experiment, which assesses the quality of explanations from the accuracy of predicting gold labels (Wiegreffe et al., 2021). Finally, we carry out human evaluations with sampled test examples. Through these three different results, our prompting approach considerably improves the humor explanation performance of three important LLMs, including one zero-shot GPT-3.5 and two finetuned T5 (Raffel et al., 2020) and BART (Lewis et al., 2020).

To summarize, our key contributions are:

1. We curate **ExFunTube**, a dataset consisting

of 10,136 user-generated, funny short-form videos. Each video is annotated with timestamps and explanations of funny moments. As compared in Table 1, our ExFunTube is unique over existing datasets in that our videos cover a wide range of domains with various types of humor that necessitate a multimodal understanding of the content.

2. We design a zero-shot video-to-text prompting that converts video content into text to maximize LLMs' ability to explain video humor.

3. With three different evaluation methods of model-based lexical scores, rationale quality scores, and human evaluations, we verify that our prompting improves LLMs' performance on humor explanation.

## 2 Related work

**Humor Understanding**. It has been a long-standing question whether AI models can understand humor in text, images, or videos. Early studies focused on classifying whether text (Annamorad-nejad and Zoghi, 2020), images (Chandrasekaran et al., 2016), or videos (Hasan et al., 2019; Castro et al., 2019; Patro et al., 2021) are humorous or not. Some studies, such as Chandrasekaran et al. (2016), also rate the degree to which abstract scenes are perceived as humorous. However, binary classifications or ratings do not fully evaluate whether a model understands humor in detail. Recent humor studies have shifted towards having models explain humor. Sun et al. (2022) augment the SemEval 2017 Task 7 (Miller et al., 2017) with funniness ratings and explanations. Hessel et al. (2022) augment the New Yorker cartoon captions with explanations. Desai et al. (2022) propose a dataset of explanations for sarcastic captions, and Kumar et al. (2022) collect sarcastic videos from a sitcom with explanations.

**Natural Language Explanation**. As tasks of interest become increasingly complex, predicting labels may not be enough to evaluate the models' true understanding. Thus, some works make models explain their decisions as an alternative. For instance, FLUTE (Chakrabarty et al., 2022) augments e-SNLI (Camburu et al., 2018) to curate figurative texts with labels for natural language inference (NLI) tasks and evaluate model-generated explanations. To evaluate model explanations, they utilize a rationale quality metric suggested by Wiegreffe

et al. (2021). As word-overlap scores may be insufficient for the evaluation of explanation, Wiegreffe et al. (2021) propose a rationale quality metric that calculates the difference of prediction scores for gold labels when rationales are provided or not: Acc (IR → O) − Acc (I → O), where I, R, and O denote input, rationale, and gold label, respectively. In addition, Sun et al. (2022) evaluate explanations by comparing the accuracy of joke classification with and without explanations: Acc (IE → O) − Acc (I → O) where E denotes explanation. We introduce a moment localization task to compute the rationale quality score of the video explanation.

**Modular Vision-Language Learning**. As pretrained models become larger and are trained with extensive datasets, various multimodal comprehension tasks have been tackled by composing these pretrained models. One approach is to transform visual information into discrete text words (Zeng et al., 2022; Yang et al., 2022; Wang et al., 2022b). Zeng et al. (2022) propose a modular framework that leverages LLM to construct the input text for the subsequent model based on the output of multimodal models in the previous stage. They demonstrate performance improvements in image captioning and visual question answering (VQA) tasks. Another approach connects pretrained models through continuous feature embeddings (Patro et al., 2021; Alayrac et al., 2022; Tiong et al., 2022). Li et al. (2023a) pretrain additional lightweight modules that bridge the frozen image encoder and LLMs to eliminate the modality gap between the two frozen pretrained models. Tewel et al. (2022) connect the frozen image encoder with the frozen language decoder and evolve additional pseudo tokens during inference time to perform the video captioning task. Recently, there have been efforts to integrate these two different approaches. Li et al. (2023b) introduce VideoChat, a chat-centric video understanding system consisting of two modules: VideoChat-Text and VideoChat-Embed. The former generates text descriptions from the video and the latter encodes the video as embeddings. These text descriptions and embeddings are combined with a received question to form a prompt, based on which the LLM generates a response.

In our work, we combine vision-language pretrained models with LLMs through text for two uses: (i) video filtering for collecting multimodal funny videos and (ii) video-to-text generation to provide LLMs with a prompt of video content.

# 3   The ExFunTube Dataset

The ExFunTube dataset comprises 10,136 videos, each annotated with timestamps of funny moments and corresponding explanations describing why each moment is humorous. The purpose of this dataset is to evaluate the models' ability to explain why a given video is funny as a measure of understanding video humor.

## 3.1   Video Collection and Filtering

We initially crawl all 220K videos shared on the subreddit "r/youtubehaiku,"[1] where people share humorous short-form YouTube videos lasting up to 30 seconds. To ensure multimodal humor in videos, we design a four-step filtering pipeline that selects videos with both visual and verbal elements contributing to humor, as shown in Figure 2.

**Video Caption and Transcript**. In the first step (Figure 2 (a)), we obtain a transcript and a video caption to describe the verbal and visual elements of a video clip, respectively. We extract a video caption using a zero-shot video captioning model (Tewel et al., 2022). Since our dataset contains diverse videos such as animations and edited videos not present in previous video datasets, we choose a model that utilizes both CLIP (Radford et al., 2021) and GPT-2 (Radford et al., 2019), which are pretrained on huge Web-sourced data. We transcribe audio from the video clip using a speech-to-text model Whisper (Radford et al., 2022). We remove videos with no speech or in languages other than English.

**Multimodal Humor**. Our goal is to collect the videos that are funny from both verbal and visual elements, instead of funny from only one modality. Thus, as shown in Figure 2 (b), we first verify that the video is verbally funny; we do this by whether GPT-3.5 can find a funny utterance given a pair of the video caption and the transcript. If GPT-3.5 detects no funny utterances, we filter out the video. Next, as shown in Figure 2 (c), we again prompt GPT-3.5 to find a funny utterance with only a transcript (*i.e.*, no video caption). If no funny utterance is detected, then we accept this video. The rationale is that the humor of this video is *multimodal*; the visual caption is required to identify the fun in the video. Otherwise, if GPT-3.5 can find a funny utterance in this case, we perform a further inspection as follows.

---

[1]https://www.reddit.com/r/youtubehaiku/

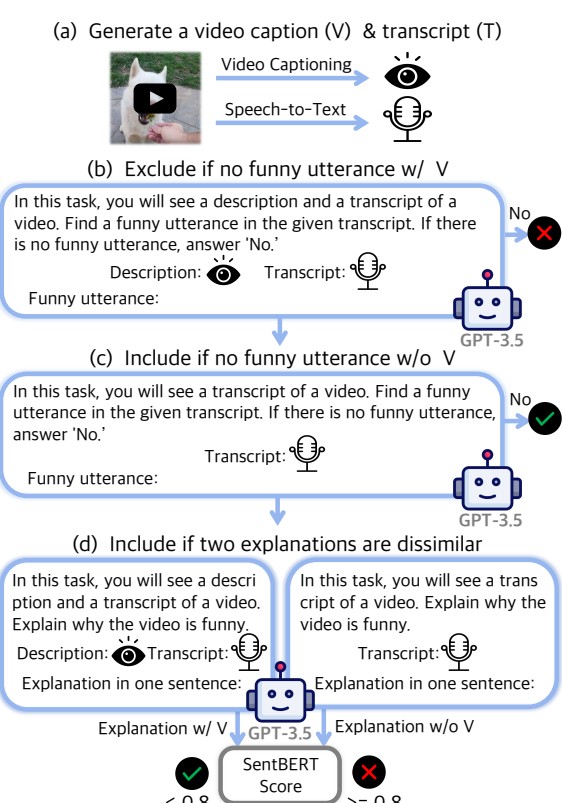

Figure 2: The video filtering pipeline selects multimodal funny videos. Red boxes display the actual prompts provided to GPT-3.5. See the details in § 3.1. (a) We generate a transcript and a caption from the input video. (b) Via GPT-3.5 prompting, we filter out the video that is not funny from the transcript and caption. (c) The video is accepted if it is funny from both the transcript and caption but not from the transcript only, since its humor is multimodal. (d) GPT-3.5 generates humor explanations with or without the video caption. We remove the videos if they are too similar since their humor is not multimodal. Examples for each case are presented in the Appendix.

**Difference in Explanations**. In the last step (Figure 2 (d)), GPT-3.5 is prompted to generate explanations in one sentence for the two cases: when given both a video caption and a transcript and when given only a transcript. We then measure the similarity between the two explanations using the SentBERT score (Reimers and Gurevych, 2019), which embeds each sentence and calculates the cosine similarity of their embeddings. The reason for adopting the SentBERT score is that it can reflect the semantics of the entire sentence. If the score is higher than the threshold, we exclude the video since the video caption does not contribute to the humor explanation. Otherwise, the video is accepted.

**Rationale of Our Pipeline**. There has yet to be a method to gauge the extent and manner in which visual elements contribute to humor. In other benchmarks, the multimodality of datasets has been validated by analyzing the performance gap when visual information is either provided or not (Hasan et al., 2019; Patro et al., 2021; Kumar et al., 2022). Similarly, we collect videos that exhibit differences in the assigned task (*i.e.*, identifying humorous utterances by GPT-3.5) with or without visual information. In the field of NLI, previous works (Liu et al., 2022; Wiegreffe et al., 2022; Chakrabarty et al., 2022) leverage the power of LLMs such as GPT-3 (Brown et al., 2020) in creating figurative language examples or explanations for them. Likewise, we use GPT-3.5 to check the difference between generated explanations. To the best of our knowledge, this is the first approach that employs explanations for curating a dataset. Thanks to the pipeline, we can collect 21K high-quality multimodal humorous videos.

**Postprocessing**. To ensure that our dataset does not contain any disrespectful or harmful content towards individuals or animals, we conduct a thorough manual review of all 21K videos. We filter out the videos using the five criteria based on the safety objectives outlined by Thoppilan et al. (2022): (i) Discrimination: videos displaying discrimination based on race, gender, sexual orientation, age, or disability. (ii) Animal cruelty: videos depicting acts of animal cruelty, such as a cat falling. (iii) Dangerous goods, services, activities, or self-harm: videos featuring dangerous content like drugs, violence, or bullying. (iv) Obscenities or profanities: videos containing explicit language or sexual actions. (v) Shocking content: videos that include shocking content, such as gunshots or explosions. After the filtering, about 50% of the videos are removed, and we are left with 10,136 videos.

### 3.2 Data annotations

We crowdsource via Amazon Mechanical Turk (AMT) to annotate start and end timestamps of funny moments and provide text explanations for each moment. To participate in our dataset annotation, workers must meet the following criteria: a HIT approval rate of 99% or higher, a total of more than 10,000 approved HITs, and be located in one of the countries of AU, CA, GB, NZ, or US. We conduct a qualification test for these workers, selecting those who can effectively explain humor.

Out of 219 workers, only 60 pass the qualification test, indicating our thorough selection.

For each video, we instruct one worker first to identify up to three funny moments within a video (up to 30 seconds long) and then annotate why each moment is funny. To make workers explain both humor elements and justifications, we provide a recommended format: *"[What is funny]. It is funny because [Why funny]"*. We only accept responses including both descriptions (*What*) and justifications (*Why*) and reject those that lack either. Given the difficulty of the task, we offer detailed feedback to the workers, helping them improve their performance with a high annotation standard.

As a result, we obtain 11,166 explanations, each paired with start and end timestamps of the moment. They consist of 44.3 words on average. Out of 10,136 videos, 9,222 contain one funny moment, 798 contain two, and 116 contain three. Most videos contain a single funny moment since videos are typically shorter than 30 seconds. However, given the varied content in each video, there can be any number of funny moments.

## 4 Approach

We explore an approach to explain video humor. Our idea is first to convert the video content into fine-grained text and then take advantage of recent powerful LLMs in a zero-shot manner. We design to extract as much information from videos into text as possible. Figure 3 shows a zero-shot video-to-text prompting that converts the video content into a text input to LLMs.

### 4.1 Fine-grained Text Prompts

Videos contain visual and audio modalities. The audio is further split into speech and sound. For each component, we initially generate text descriptions using state-of-the-art zero-shot models. Then, we arrange text descriptions in chronological order and use them as a prompt.

**Visual**. In order to populate high-quality text descriptions about the visual, we first (i) segment the video, (ii) generate multiple frame captions, and (iii) retrieve the best-matching caption with the video-to-text model.

First, we employ PySceneDetect[2] to divide a video into a set of $N$ segments based on visual changes. During the filtering pipeline (§3.1), the speech-to-text model Whisper generates timestamps

---

[2] https://github.com/Breakthrough/PySceneDetect

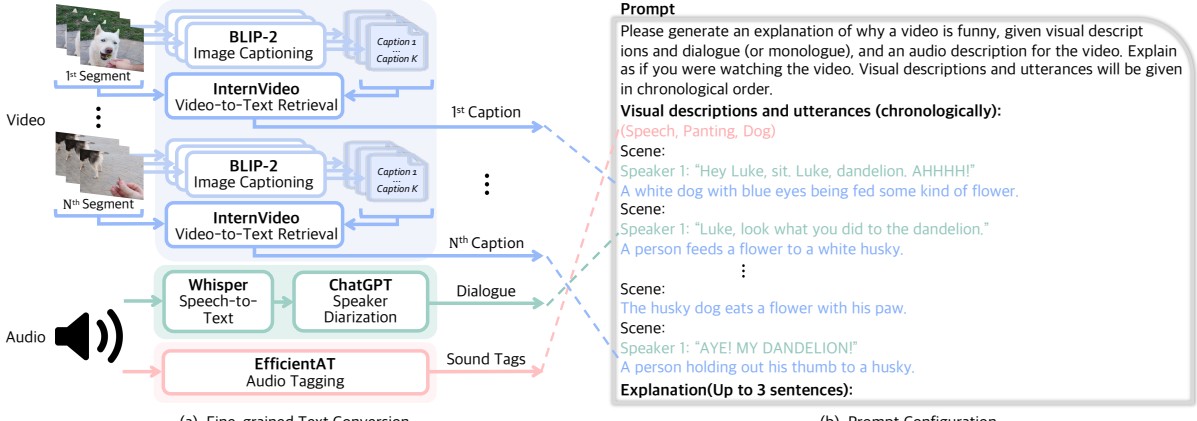

Figure 3: (a) A zero-shot video-to-text prompting for converting video content into fine-grained text (§ 4.1). For the visual modality, the video is first divided into $N$ segments, for each of which many possible captions are generated, and the best one is chosen finally. For audio modality, a transcript with speaker separation and sound tags are obtained. (b) The fine-grained text is configured as an input prompt to LLMs (§ 4.2).

for each utterance. We also use them to split the segments further, resulting in more fine-grained and semantically meaningful video segments.

Next, we extract frames at a rate of 5fps from each of the $N$ video segments. We generate $K (= 20)$ captions per frame using the image captioning model BLIP-2 (Li et al., 2023a) with a "Who is doing what?" prompt, which can enhance action detection. We then have a frame caption corpus (# Frames $\times K$ captions) per segment. Subsequently, we use the video-to-text model InternVideo (Wang et al., 2022a) to retrieve the caption that best matches each video segment from the respective frame corpus. Finally, we obtain one caption per segment, resulting in a total of $N$ captions, which are fine-grained descriptions of the visual component.

**Speech**. We transcribe audio with Whisper (Radford et al., 2022) as done in our video filtering pipeline. We then predict the number of speakers and assign speakers to each utterance utilizing ChatGPT (OpenAI, 2023). This speaker separation helps a deep understanding of dialogue.

**Sound**. We extract sound tags to provide more context. We use an audio tagging model (Schmid et al., 2022) to classify the entire audio stream. We select the top 3 predicted tags that have a higher confidence value than the threshold (0.3). We concatenate the tags and insert them at the beginning of the prompt. This can provide the model with an overall atmosphere of the video.

## 4.2 Prompt Configuration and LLMs

After extracting text from visual, speech, and sound, we configure the prompt like an example of Figure 3. The prompt starts with a predefined text "Please generate ~" to instruct LLMs to explain as if they are watching the video. We then include sound tags enclosed in parentheses and arrange the extracted text of speech and visuals for each video segment chronologically. To distinguish between video segments, we begin each segment with "Scene: ". Finally, we ask LLMs to generate an explanation of up to three sentences.

**LLMs**. Although any LLMs can be adopted, we use three different ones: finetuned T5 (Raffel et al., 2020) and BART (Lewis et al., 2020), and zero-shot GPT-3.5 text-davinci-003.

## 5 Experiments

We experiment with different models to see how well they explain the humor in the ExFunTube videos. We evaluate the models in three different ways of model-based automatic scores, rationale quality experiments, and human evaluation.

## 5.1 Experimental Setup

**Baselines**. We evaluate four types of explanation models. (i) **Text-only LLMs** generate explanations when only a transcript is provided (*i.e.*, no use of visual). We use T5 Large and BART Large with finetuning and GPT-3.5 as a zero-shot model. (ii)

| | | Automatic Score | | | | | | | | Rationale Quality Score (↓) | | Human Evaluation (↑) |
| | | SentBERT (↑) | | | | | ROSCOE (RA) (↑) | | | | | |
| | | @0.7 | @0.6 | @0.5 | @0.4 | Mean | @0.8 | @0.7 | Mean | @0.3 | @0.5 | Rating |
|---|---|---|---|---|---|---|---|---|---|---|---|---|
| Text-Only | T5 | 0.154 | 0.355 | 0.585 | 0.795 | 0.534 | 0.406 | 0.871 | 0.780 | 10.3 | 21.9 | - |
| | BART | 0.169 | 0.388 | 0.617 | 0.807 | 0.545 | 0.440 | 0.875 | 0.785 | 13.7 | 30.1 | 0.178 |
| | GPT-3.5 | 0.149 | 0.310 | 0.556 | 0.774 | 0.529 | 0.371 | 0.841 | 0.772 | 18.8 | 22.5 | 0.385 |
| MAF | - | 0.149 | 0.375 | 0.604 | 0.809 | 0.541 | 0.438 | 0.880 | 0.785 | 13.1 | 25.3 | 0.131 |
| VideoChat-Text | GPT-3.5 | 0.115 | 0.345 | 0.618 | 0.839 | 0.539 | 0.414 | 0.900 | 0.783 | 13.9 | 26.5 | - |
| Our Prompting | T5 | 0.230 | 0.483 | 0.719 | 0.887 | 0.584 | 0.543 | 0.932 | 0.804 | **2.9** | 12.5 | - |
| | BART | **0.238** | 0.500 | 0.730 | 0.886 | 0.588 | 0.554 | 0.935 | 0.805 | 6.3 | 23.9 | 0.282 |
| | GPT-3.5 | 0.214 | **0.541** | **0.806** | **0.945** | **0.602** | **0.639** | **0.971** | **0.817** | 5.5 | **9.3** | **0.523** |
| Gold | - | - | - | - | - | - | - | - | - | - | - | 0.792 |

Table 2: Humor explanation results in terms of automatic scores (SentBERT and ROSCOE), rationale quality scores, and human rating. In the automatic scores, @K shows the proportion of test explanations of which scores are higher than K, and the mean column is the average score of each metric. For rationale quality scores with funny moment localization, we adopt two IoU thresholds, 0.3 and 0.5; lower scores are better. For human rating, five workers rate each of 100 randomly selected test videos from No (0), Weak No (0.25), Neutral (0.5), Weak Yes (0.75), to Yes (1). After excluding the highest and lowest scores, the remaining scores are averaged.

**MAF** (Kumar et al., 2022) is a multimodal end-to-end model designed for video sarcasm explanation. It generates explanations by receiving features of the three components (visual, speech, and audio). We train the model on our dataset. (iii) **VideoChat-Text** (Li et al., 2023b) is a multimodal prompting framework that textualizes video information into text, including video/clip captions, objects contained in the video and a transcript. Given the prompt, GPT-3.5 generates explanations in a zero-shot manner. (iv) **LLMs with our prompting** generate explanations given a prompt created by our zero-shot video-to-text prompting, using the same LLMs as (i) of T5, BART, and GPT-3.5. Note that T5 and BART models are finetuned to generate explanations given generated prompts, while GPT-3.5 generates in a zero-shot manner.

**Explanation Generation**. For all finetuned models on our dataset, we employ K-fold cross-validation as follows. We divide the entire dataset of 10,136 videos into five equal-sized subsets. In each iteration, we train the model on three subsets, use one subset for validation, and test on the remaining subset. We repeat this process five times, rotating the test subset in each iteration. Finally, we obtain predicted explanations for the entire set.

**Evaluation**. To compare the predicted explanation with the gold explanation for each video, we concatenate explanations for each moment into a single, unified explanation. For more details on experiments, please refer to the Appendix.

## 5.2 Results of Model-based Automatic Scores

Since the metrics based on word overlaps may fail to reflect faithfulness and plausibility as highlighted by Sun et al. (2022), we evaluate explanations using two model-based scores: SentBERT Score and ROSCOE (Golovneva et al., 2022). ROSCOE is a suite of metrics designed to evaluate the reasoning process within a chain-of-thought prompting (Wei et al., 2022). It is suitable for our explanation tasks since our goal is to uncover the *reason* for laughter (*i.e.*, why is the video humorous?) Among the various scores provided by ROSCOE, we use the reasoning alignment (RA) score, which computes the contextual similarity between the hypothesis and reasoning.

Table 2 reports the model-based automatic scores of different methods. We show not only the mean metric values but also the proportions of the test set with scores higher than various thresholds; @$K$ represents the proportion of data points with scores equal to or greater than $K$.

The results show that, except for SentBERT @0.7, GPT-3.5 with our prompting reaches the best performance. Especially, the SentBERT and ROSCOE scores with our prompting are higher than those with text-only baselines in all cases. In addition, our method outperforms the multimodal end-to-end baseline MAF and the multimodal zero-shot prompting baseline VideoChat-Text. The comparison of @$K$ metrics shows even more significant differences, particularly for SentBERT @0.5 and

ROSCOE @0.8, where the performance margin ranges from 0.1 (BART) to 0.27 (GPT-3.5) compared to the text-only baselines. This means that using transcripts alone may not be sufficient to understand the humor in our videos.

## 5.3 Results of Rationale Quality Scores

We conduct a rationale quality experiment following Wiegreffe et al. (2021) and Sun et al. (2022). Since our dataset consists of videos, unlike theirs, we adapt the experimentation scheme by evaluating the rationale quality through a moment localization task, which aims at predicting funny moments defined by their start and end timestamps in a video given the text explanation.

We use QD-DETR (Moon et al., 2023) as a localizer and divide the entire dataset into 8:1:1 splits for training (8,110), validation (1,013), and testing (1,013). During the training, the localizer is learned to predict the gold timestamp given a gold explanation. At inference, we compute the rationale quality as the prediction difference of the localizer between when given a model-generated explanation and when given a gold explanation.

Let $M$ be a model-generated explanation, $G$ be a gold explanation, and $\tau$ be a threshold. For each test data point, we calculate the maximum IoU from the top 5 candidates given $M$ or $G$, respectively denoted as $\text{IoU}_M$ or $\text{IoU}_G$. We use the top 5 since there can be at most three funny moments in a single video and the localization predictions can overlap with each other. We compute the difference when $\text{IoU}_M > \tau$. The final score $S$ is the sum of differences for all test data:

$$S = \sum_{i=1}^{n} (\text{IoU}_{G_i} - \text{IoU}_{M_i}) \cdot \mathbb{1}(\text{IoU}_{M_i} > \tau),$$

where $n$ is the number of test data points, and $\mathbb{1}(\cdot)$ is the indicator function.

Table 2 shows the results when the IoU threshold $\tau$ is set to 0.3 and 0.5. A lower score is better as it is closer to the gold standard. In each LLM, the performance improves when our prompting is included compared to corresponding text-only ones. In particular, our approach improves GPT-3.5 the most, with the threshold at 0.3 resulting in a score gap of 13.3, and at 0.5, a score gap of 13.2. Again, the performance of all LLMs with our prompting is better than MAF and VideoChat-Text.

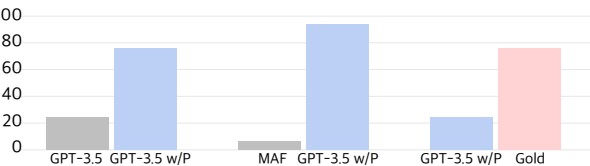

Figure 4: Results of human preference: comparing GPT-3.5 with our prompting to text-only GPT-3.5, MAF, and Gold, respectively.

## 5.4 Results of Human Evaluations

For human evaluation, we employ 10 AMT workers using the same criteria as in the dataset annotation but excluding the ones who already participated in the annotation. We randomly select 100 videos and evaluate explanations generated by all models except baselines using T5 and VideoChat-Text, which show worse automatic scores than other text-only or multimodal baselines. We obtain human evaluations with two methods: rating and comparison.

For the rating, workers are asked to rate each explanation according to No (0), Weak No (0.25), Neutral (0.5), Weak Yes (0.75), and Yes (1) and check any shortcomings. We ask five workers for each explanation, exclude the highest and lowest scores, and take the average. For the comparison, workers compare GPT-3.5 with our prompting to (1) Text-only GPT-3.5, (2) MAF, and (3) Gold explanations and choose the better explanation. We ask five workers for each pair of comparisons.

The rating results are presented on the far right of Table 2. The scores of BART and GPT-3.5 increase by about 0.1 when our prompting is included. The comparison results are presented in Figure 4. The number of votes for text-only GPT-3.5 is significantly lower than that of GPT-3.5 with our prompting, indicating that visual information is valuable, and our prompting helps convey visual information effectively. In both rating and comparison, MAF shows lower performance than the text-only models despite being a multimodal model. This suggests that providing visual information as text to LLMs could be more effective than training the multimodal model end-to-end. Moreover, GPT-3.5 with our prompting, which shows the best results, still scores lower than Gold, indicating that understanding and explaining the humor in our dataset still remains unsolved.

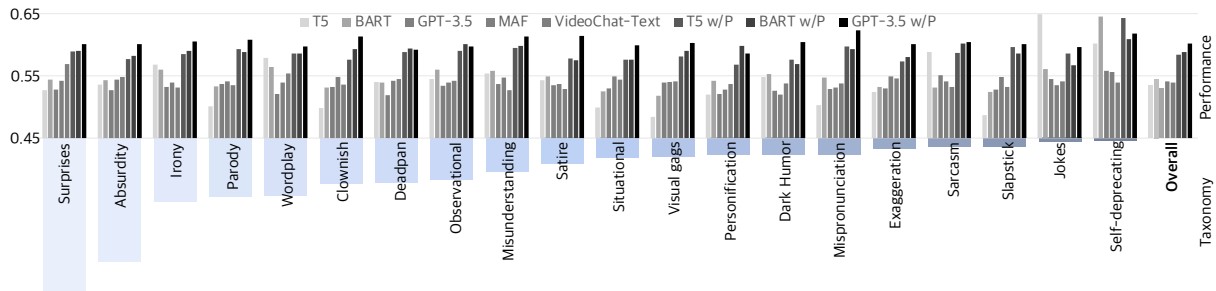

Figure 5: Explanation performance according to humor taxonomy. We categorize all videos into 20 humor classes and compare the performance of eight different baselines in terms of the SentBERT score. The humor taxonomy is arranged in descending order of proportion in our dataset.

## 5.5 Analyzing LLMs with Humor Taxonomy

We classify our dataset into a total of 20 humor categories referring to Martin and Ford (2018) and Buijzen and Valkenburg (2004), and observe the performance of baselines by the humor taxonomy. We provide ChatGPT with 20 categories along with a brief description and one example (*i.e.*, one-shot learning) and instruct ChatGPT to classify the video based on the given explanation. Thanks to ChatGPT's powerful in-context learning capability, we effectively classify 10,136 videos based on their corresponding explanations.

Figure 5 shows the models' performance by humor categories. Excluding the Jokes and Self-deprecating classes, the performance increases with our prompting in all categories. In particular, the performance significantly increases in Clownish humor, Visual gags, and Slapsticks, which heavily reflect visual elements. This indicates that our zero-shot video-to-text prompting effectively conveys visual elements to the LLM.

## 5.6 Ablation Study

We compare the importance of each modality in humor explanation. Table 3 presents the results of SentBERT and ROSCOE scores when visual, speech, and sound components are not included in the prompt one by one. In GPT-3.5 with our prompting, the performance without the visual component drops as much as when the speech is removed, indicating that the visual component plays an important role in our dataset. Moreover, the performance decreases when either of the components is removed,

|  | GPT-3.5 w/ Prompting | | | |
|---|---|---|---|---|
|  | w/o V | w/o T | w/o A | w/ V, T, A |
| SentBERT | 0.512 | 0.497 | 0.574 | **0.602** |
| ROSCOE (RA) | 0.778 | 0.763 | 0.801 | **0.817** |

Table 3: Ablation results of GPT-3.5 with our prompting measured by SentBERT and ROSCOE scores when each modality component is removed. V, T, and A denote visual, speech, and sound, respectively.

which suggests that all three components are crucial for understanding and explaining humorous videos in our dataset. Additional ablation studies are presented in the Appendix.

## 6 Conclusion

We introduced ExFunTube, a dataset consisting of 10,136 user-generated videos annotated with timestamps and explanations of funny moments. Our dataset aims to assess how well AI models understand and explain video humor. We devised a zero-shot video-to-text prompting to make existing LLMs better explain the video content. With three different evaluation methods, we demonstrated that the humor in our dataset is multimodal, and our prompting maximized LLMs' ability to generate explanations.

However, as the performance still falls short of human levels, our dataset remains sufficiently challenging and calls for future research. Furthermore, we can consider the training of the model using user feedback for personalized humor understanding.

## Limitations

Since the copyright remains with the original owners of our dataset videos, we will only distribute URLs instead of videos.

Our method relies on the performance of existing state-of-the-art models, as we used them in a zero-shot composition. Also, our approach composes models through text, so it could also be explorable to use an adaptor-based method for prompt tuning during inference.

We measured the videos by dividing them into three modalities, but we did not consider the temporal information of sound. As timing can play a role in humor, analyzing the sound in accordance with the timeline could be helpful.

Lastly, humor is subjective, which means that our collected explanations may be subjective, too.

## Ethics Statement

We put much effort into ensuring that our dataset contains no inappropriate videos that may raise ethical issues. Based on the safety rules of Thoppilan et al. (2022), authors manually viewed each video entirely from start to end and filtered the video if there was any content that corresponded to the filtering criteria presented in the dataset postprocessing. Although we carefully reviewed all the videos, there could still be some videos that are not comfortable for someone. If such inappropriate videos are found, we will remove them in the future. Also, since we only recruit workers in AU, CA, GB, NZ, and US as mentioned in the Appendix, the cultural and geographic biases may influence humor explanations.

## Acknowledgments

We sincerely thank Jaekyeom Kim, Jaewoo Ahn, Soochan Lee, Wonkwang Lee, Yeda Song, and Jaehyeon Son for their valuable comments. We would also like to thank AMT workers for their commitment to building the **ExFunTube** dataset. This work was supported by the SNU-Global Excellence Research Center establishment project, Basic Science Research Program through the National Research Foundation of Korea (NRF) funded by the Ministry of Education (RS-2023-00274280), Institute of Information & communications Technology Planning & Evaluation (IITP) grant funded by the Korea government (MSIT) (No. 2022-0-00156, Fundamental research on continual meta-learning for quality enhancement of casual videos and their 3D metaverse transformation), and Institute of Information & communications Technology Planning & Evaluation (IITP) grant funded by the Korea government (MSIT) (No. 2021-0-01343, Artificial Intelligence Graduate School Program (Seoul National University)). Gunhee Kim is the corresponding author.

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

## A  Experimental Details

**Video Filtering Pipeline**. In the video filtering pipeline, we utilize a zero-shot video captioning model from Tewel et al. (2022), a speech-to-text model Whisper (Radford et al., 2022), and GPT-3.5 (Ouyang et al., 2022). For the video captioning model, we optimize pseudo tokens for 25 iterations at inference time to guide the pretrained GPT-2 (Radford et al., 2019) with the CLIP ViT-L/14 image encoder (Radford et al., 2021). We use AdamW optimizer (Loshchilov and Hutter, 2019) with a learning rate of 0.008 and an L2 weight decay of 0.003. For Whisper, we use the large-v2 model. For GPT-3.5, we use text-davinci-003 and set the temperature to 0 for funny utterance detection and 0.3 for explanation generation.

  **Video-to-Text Prompting**. During the prompting stage, we use BLIP-2 (Li et al., 2023a), InternVideo (Wang et al., 2022a), Whisper, ChatGPT (OpenAI, 2023), and an audio-tagging model from Schmid et al. (2022). We use the coco-pretrained BLIP-2 model with nucleus sampling. For InternVideo, we use CLIP ViT-L/14 as the image encoder. We set the temperature to 0.3 for ChatGPT, and we use the mn40_as model for audio tagging.

  **Explanation Generation**. To generate explanations with baseline models, we finetune T5 (Raffel et al., 2020) and BART (Lewis et al., 2020) with a batch size of 4 for 5 epochs. We use the AdamW optimizer with a learning rate of 2e-5 and an L2 weight decay of 0.01. Additionally, we train MAF (Kumar et al., 2022), a multimodal end-to-end model with an adaptor added to BART, with a batch size of 4 for 20 epochs. We use the AdamW optimizer with an L2 weight decay of 1e-4, and the learning rate is set to 5e-8 for BART parameters and 5e-7 for the remaining parameters. We use BART Large for all models.

  **Rationale Quality Experiments**. For the rationale quality experiments with moment localization, we train QD-DETR (Moon et al., 2023) with a batch size of 128 for 200 epochs. We use the AdamW optimizer with a learning rate of 1e-5 and an L2 weight decay of 1e-4. We optimize with the moment retrieval loss consisting of the L1 loss, the cross-entropy loss and the generalized IoU loss. We use the loss balancing terms of 10, 1 and 2 for each of them, respectively. We do not use the saliency loss. We use the bert-base-uncased model (Devlin et al., 2019) as the text encoder with the max query length set to 400 and CLIP ViT-L/14 as the video

|  |  | SentBERT | ROSCOE (RA) |
|---|---|---|---|
| T5 w/ Prompting | w/o V | 0.540 | 0.783 |
|  | w/o T | 0.463 | 0.753 |
|  | w/o A | 0.578 | 0.801 |
|  | w/ V, T, A | **0.584** | **0.804** |
| BART w/ Prompting | w/o V | 0.551 | 0.788 |
|  | w/o T | 0.497 | 0.767 |
|  | w/o A | 0.587 | 0.805 |
|  | w/ V, T, A | **0.588** | **0.805** |

Table 4: Ablation results of T5 and BART with our prompting measured by SentBERT and ROSCOE scores when each modality component is removed. V, T, and A denote visual, speech, and sound, respectively.

encoder. We sample video frames at a rate of 1 fps.

  Except for the aforementioned hyperparameters, we use the default values for all models.

## B  Additional Ablation Study

We conduct ablation experiments on BART and T5 with our prompting as well, and the results are as shown in Table 4. Similar to the results of GPT-3.5 with our prompting, using all modalities achieves the best performance, and there is a certain degree of performance decrease when the visual component is removed.

## C  Crowdsourcing Details

We use three different user interfaces of Amazon Mechanical Turk (AMT) for (i) annotating the timestamps and explanations of funny moments, and the human evaluation of (ii) rating and (iii) comparison, as shown in Figures 6-8, respectively. We guarantee AMT workers receive fair wages of approximately $18 per hour. Additionally, we allocate about $2 as compensation for each data point and grant additional wages to workers contributing extended time and effort.

## D  Case Study

Figures 9-12 show representative videos accepted or excluded by our video filtering pipeline. Figures 13-18 provide several examples to demonstrate humor explanations that our baseline models actually generate. We color-code relevant (blue) and irrelevant (red) information contained in generated explanations. LLMs with our prompting, especially GPT-3.5, correctly explain the funny moments in Figures 13-16 while text-only LLMs and MAF fail to. All the models fail to explain humorous moments in Figures 17-18.

## Temporal localization & Explanation

Please enter the shortest possible time range between 1 second and 5 seconds, and explain **why the video is funny** including **visual information** and **funny remarks** in the speech. If there are more than one moments, please add an input field and **enter them individually**. Filling fields a lot isn't necessarily a good thing. Write only what makes you laugh particularly.

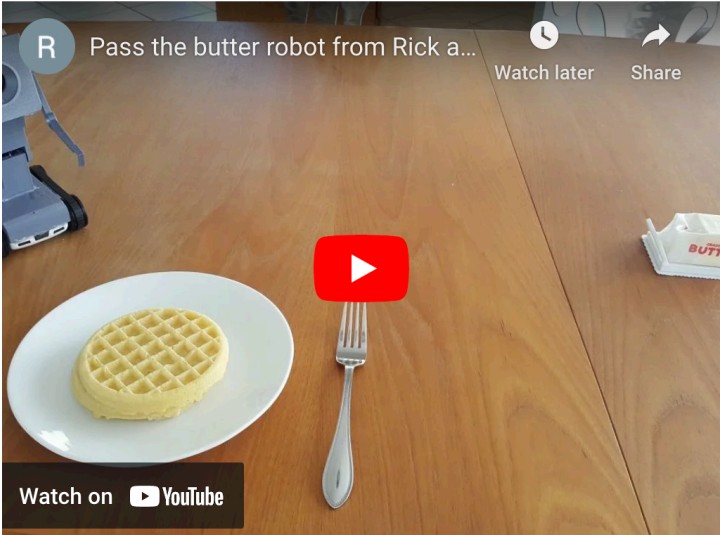

For your convenience, we provide video subtitles. This could be wrong.

| Video subtitles |
| --- |
| What is my purpose? Pass the butter. Thank you. What is my purpose?. You passed butter. Oh my god. |

Start time (secon⋯     End time (seconds)     Explanation
20                     24                     The robot realizes his entire purpose in 'life' is to "pass butter" and reacts with a verbal "oh my God" while slumping in dismay. It's funny, because it's a reaction we'd normally attribute to a human and not a robot.

Start time (s⋯        End time (se⋯          Explanation

Add field +

If your answer is not being submitted, please check the requirements below.
1. Start time and end time entered must be less than length of the video.
2. Start time must be less than end time.
3. Enter the time range as short as possible. (The maximum limit was removed by reflecting the feedback.)
4. The explanation must be **at least 100 characters.**

(Optional) Please let us know if anything was unclear, if you experienced any issues, or if you have any feedback for us.

Submit

Figure 6: A user interface for annotating timestamps and explanations of humorous moments. Workers are asked to watch a video, identify up to three funny moments, and provide the start/end timestamps along with the explanation for each moment.

## [Your Work]

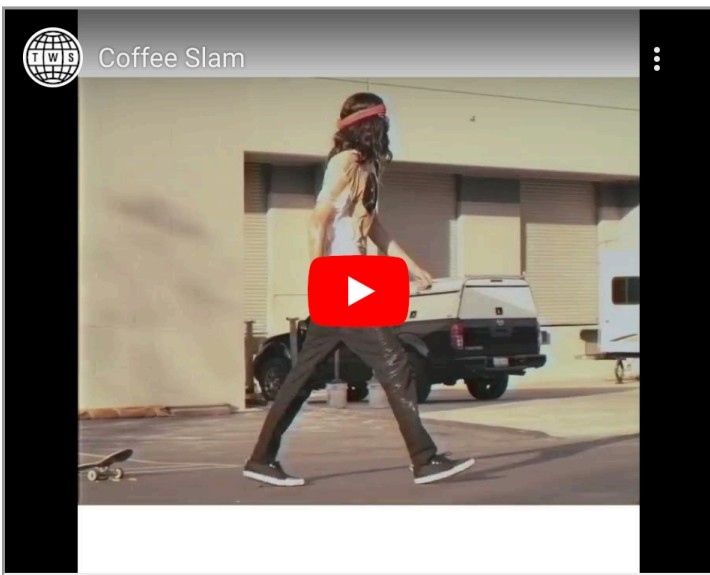

### Rate Explanations

Please read each explanation and rate it as "No", "Weak No", "Neutral", "Weak Yes", or "Yes" based on the criterion and then check shortcomings of each explanation. More details on the shortcomings are provided below.

#### Descriptions of Shortcomings

- **Missed Humor** : The explanation fails to identify the key aspect of the humor in the video.
- **Too Vague** : The explanation isn't specific about the funny parts of the video.
- **Incorrect Details** : The explanation has wrong information or misinterprets the funny elements.
- **Lacking Context** : The explanation misses important context or background information.
- **Too Verbose** : The explanation is overly lengthy, making it harder to understand the humor.

---

#### Explanation #1.

> The delivery of the line "I got the coffees!" is delivered with an exaggerated enthusiasm, which is funny in itself. The response of "Ah!" is delivered with a surprised and relieved expression, which adds to the comedic effect. The combination of the two lines creates a humorous moment that elicits a laugh from the viewer.

Q1. Does the explanation **clearly** describe the reasons why it is funny **including video content**?

⦿ No    ◯ Weak No    ◯ Neutral    ◯ Weak Yes    ◯ Yes

Q2. What are the shortcomings of the explanation? (Check all.)

☑ Missed Humor    ☑ Too Vague    ☐ Incorrect Details    ☑ Lacking Context    ☐ Too Verbose

☐ None

Figure 7: A user interface for human evaluation through rating. Workers are asked to rate the explanation on a scale of No, Weak No, Neutral, Weak Yes, to Yes, and to choose any shortcomings if present.

# [Your Work]

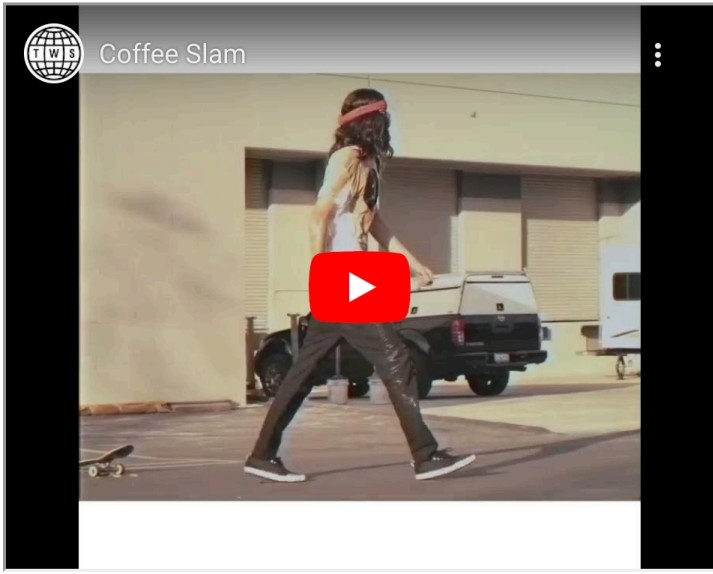

## Choose a Better Explanation

Please read the following three pairs of explanations and for each pair choose the one that explains better why the video is funny.

**Explanation #1.**

> The delivery of the line "I got the coffees!" is delivered with an exaggerated enthusiasm, which is funny in itself. The response of "Ah!" is delivered with a surprised and relieved expression, which adds to the comedic effect. The combination of the two lines creates a humorous moment that elicits a laugh from the viewer.

**Explanation #2.**

> The video starts with a man on a skateboard holding onto a pile of cups, and then the audio caption of a skateboard breaking in the background of a speech is heard. The man then falls to the ground by the trash cans, and the next scene shows a guy on a skateboard wearing a bandana. The last scene shows the same skateboarder walking next to a truck, which implies that he was the one who broke the skateboard.

Q. Which one provides more **clear** explanation of why it is funny, **including video content**?

○ Exp #1    ◉ Exp #2

Figure 8: A user interface for human evaluation through comparison. Workers are asked to compare GPT-3.5 with our prompting to text-only GPT-3.5, MAF, and Gold, respectively, and select the superior one.

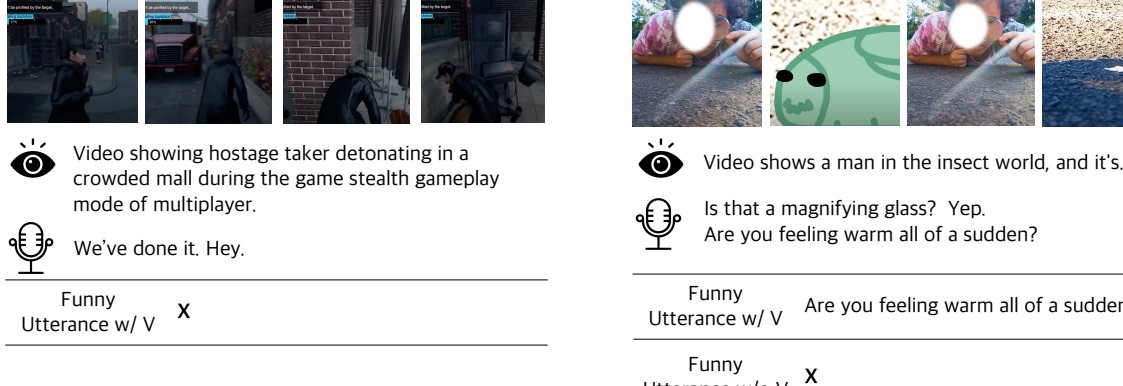

👁 Video showing hostage taker detonating in a crowded mall during the game stealth gameplay mode of multiplayer.

🎤 We've done it. Hey.

| Funny Utterance w/ V | X |
|---|---|

Figure 9: An example of a video excluded in the second step (Figure 2 (b)) of the filtering pipeline.

👁 Video shows a man in the insect world, and it's.

🎤 Is that a magnifying glass? Yep.
Are you feeling warm all of a sudden?

| Funny Utterance w/ V | Are you feeling warm all of a sudden? |
|---|---|
| Funny Utterance w/o V | X |

Figure 11: An example of a video accepted in the third step (Figure 2 (c)) of the filtering pipeline.

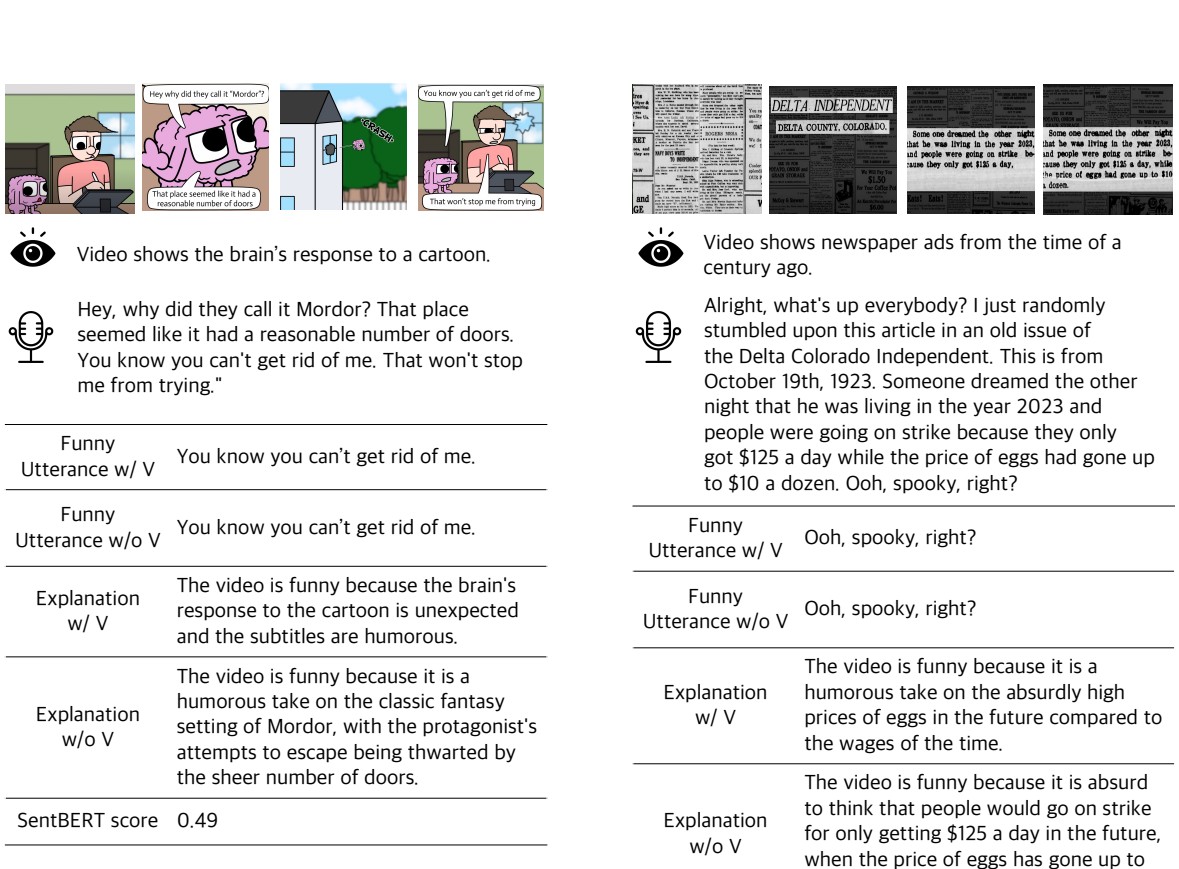

👁 Video shows the brain's response to a cartoon.

🎤 Hey, why did they call it Mordor? That place seemed like it had a reasonable number of doors. You know you can't get rid of me. That won't stop me from trying."

| Funny Utterance w/ V | You know you can't get rid of me. |
|---|---|
| Funny Utterance w/o V | You know you can't get rid of me. |
| Explanation w/ V | The video is funny because the brain's response to the cartoon is unexpected and the subtitles are humorous. |
| Explanation w/o V | The video is funny because it is a humorous take on the classic fantasy setting of Mordor, with the protagonist's attempts to escape being thwarted by the sheer number of doors. |
| SentBERT score | 0.49 |

Figure 10: An example of a video accepted in the fourth step (Figure 2 (d)) of the filtering pipeline.

👁 Video shows newspaper ads from the time of a century ago.

🎤 Alright, what's up everybody? I just randomly stumbled upon this article in an old issue of the Delta Colorado Independent. This is from October 19th, 1923. Someone dreamed the other night that he was living in the year 2023 and people were going on strike because they only got $125 a day while the price of eggs had gone up to $10 a dozen. Ooh, spooky, right?

| Funny Utterance w/ V | Ooh, spooky, right? |
|---|---|
| Funny Utterance w/o V | Ooh, spooky, right? |
| Explanation w/ V | The video is funny because it is a humorous take on the absurdly high prices of eggs in the future compared to the wages of the time. |
| Explanation w/o V | The video is funny because it is absurd to think that people would go on strike for only getting $125 a day in the future, when the price of eggs has gone up to $10 a dozen. |
| SentBERT score | 0.83 |

Figure 12: An example of a video excluded in the fourth step (Figure 2 (d)) of the filtering pipeline.

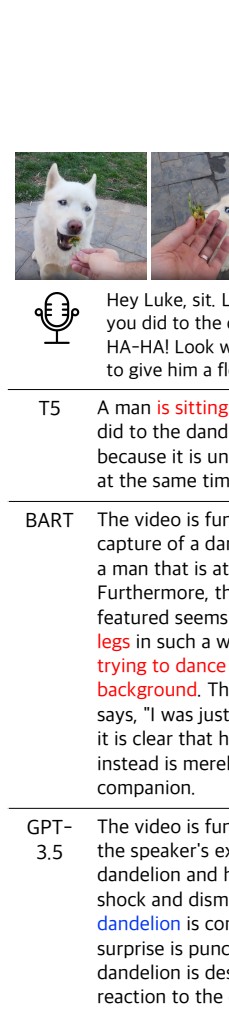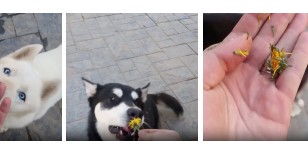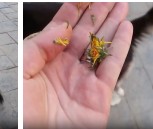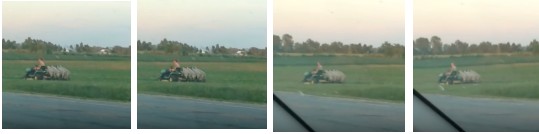

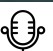 Hey Luke, sit. Luke, dandelion. AHHH! Luke, look what you did to the dandelion. Gus, look dandelion. AH-HA-HA-HA! Look what he did to the dandelion. I was trying to give him a flower. AYE! MY DANDELION!

| | |
|---|---|
| T5 | A man is sitting in a chair and he says "look what you did to the dandelion" and then he screams. It is funny because it is unexpected and because he is screaming at the same time. |
| BART | The video is funny because it presents a close up capture of a dandelion that has been knocked over by a man that is attempting to give it a flower. Furthermore, this scene is comical because the man featured seems to be comically flailing his arms and legs in such a way that makes it appear as if he is trying to dance to the beat of the song playing in the background. The video is also funny when the man says, "I was just trying to give him a flower," because it is clear that he is not in the mood to dance, but instead is merely trying to get the attention of his companion. |
| GPT-3.5 | The video is funny because of the contrast between the speaker's expectation of Luke's reaction to the dandelion and his actual reaction. The speaker's shock and dismay at Luke's destruction of the dandelion is comically exaggerated, and the speaker's surprise is punctuated by the loud "Aye!" when the dandelion is destroyed. The speaker's over-the-top reaction to the destruction of the dandelion is what makes the video funny. |
| MAF | The video is funny because it presents a clip of a man attempting to give a dandelion to another man, but instead of giving him a flower, the man throws it to the ground, which is unexpected. Furthermore, it is comical because it is unexpected to see a man throw a flower at another man. |
| T5 w/ P | A man is holding dandelions and a dog is shown eating them. It is funny because the dog is so aggressive and the man is trying to get him to sit, but he is not doing so well. |
| BART w/ P | A man gives his dog a dandelion and the dog starts licking it. It is funny because it is unexpected, and also because the dog seems to enjoy it. |
| GPT-3.5 w/ P | The video is funny because the white dog, Luke, is initially obedient when told to sit, but then proceeds to eat the dandelion instead of just sniffing it. This causes another person to scream in shock, leading to a chain reaction of laughter and more dogs being fed dandelions. The unexpected behavior of the dogs and the exaggerated reactions of the people make the video humorous. |
| **Gold** | It's funny because the white dog is shown a dandelion and then the dog eats the dandelion unexpectedly. It's funny because the black and white dog is shown a dandelion and does the same thing as the dog and eats the dandelion. Also, the man's intention was simply to give the dog a flower, not for the dog to eat it. The dog turns and notices the dandelion, then goes over and eats the dandelion from the man's hand. It's funny because of the man's exaggerated reaction. |

Figure 13: **(Correct)** An example of explanation generation results. GPT-3.5 with our prompting correctly describes the unexpected behavior of dogs and the exaggeration of the people that provoke laughter.

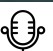 Get the muscles developed. And the muscles. She pushed a lot more.

| | |
|---|---|
| T5 | The video is funny because it presents a comical moment where a man is attempting to push his body to the point where he is unable to do so. Furthermore, the man's remark is comical because he is attempting to make a joke about how hard he is pushing his body to the point where he is unable to do so. |
| BART | The video is funny because it presents a close up capture of a female individual that appears to be attempting to "push a lot more." Furthermore, this scene is comical because it is clear that the individual featured is attempting to demonstrate a "push-a-thon" in order to "get the muscles developed," which is why it is laughable to see that the woman featured is actually attempting to push herself to the extent that she is actually "pushing" herself. |
| GPT-3.5 | The speaker's enthusiasm and repetition of the phrase "muscles" is comical, as if they are trying to emphasize the importance of the muscles. The speaker's exaggerated emphasis on the word "pushed" also adds to the humor as it implies that the person they are referring to is putting in a lot of effort. The speaker's enthusiasm and exaggeration makes the video funny. |
| MAF | The video is funny because it presents a clip of a man saying, "She pushed a lot more." It is comical because it is unexpected to see a man say this, and also because the video is edited to make it seem as if the man is implying that the woman pushed him harder than she actually did. |
| T5 w/ P | A man is riding a lawnmower and he is talking about getting his muscles developed, and then he says "she pushed a lot more." It is funny because he is talking about pushing his muscles, and also because he is talking about pushing his muscles. |
| BART w/ P | A man is riding a lawnmower next to a bunch of sheep, and he says, "She pushed a lot more." It is funny because it is unexpected to see someone push a mower like that, and also because the man says it in a way that sounds like he is bragging. |
| GPT-3.5 w/ P | The visual of a person riding a lawnmower with a herd of sheep is humorous in itself. The dialogue between the two speakers adds to the comedic effect as they discuss the man's muscles and how much he has pushed. The absurdity of the situation makes it even funnier. |
| **Gold** | It's funny to me because the implication is that lawnmower is sheep powered, because the sheep are pushing it, instead of horse powered. It's also weird to see sheep doing that. |

Figure 14: **(Correct)** An example of explanation generation results. LLMs with our prompting describe the verbal ("She pushed a lot more") and visual elements ("a person riding a lawnmower with a herd of sheep") contributing to humor.

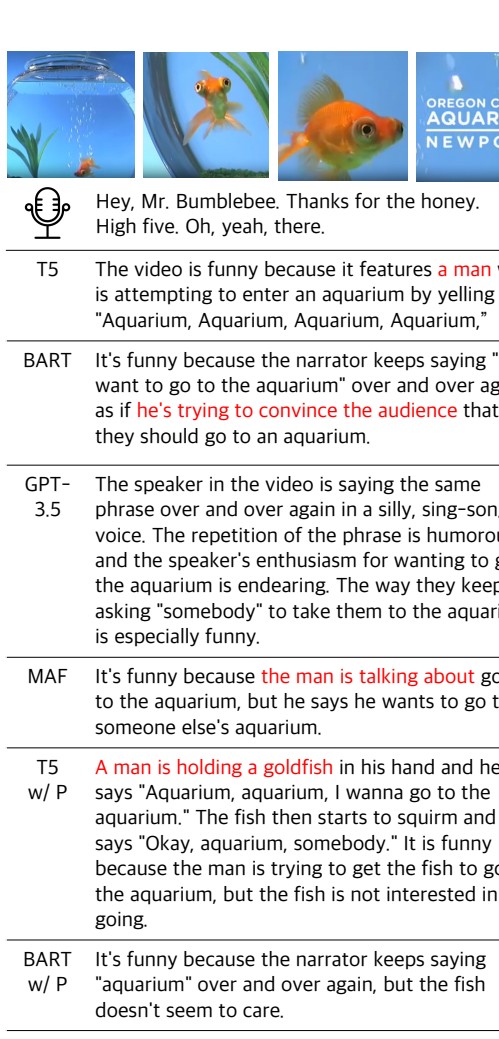

🎙️ Hey, Mr. Bumblebee. Thanks for the honey. High five. Oh, yeah, there.

| | |
|---|---|
| **T5** | The video is funny because it features a man who is attempting to enter an aquarium by yelling "Aquarium, Aquarium, Aquarium, Aquarium," |
| **BART** | It's funny because the narrator keeps saying "I want to go to the aquarium" over and over again, as if he's trying to convince the audience that they should go to an aquarium. |
| **GPT-3.5** | The speaker in the video is saying the same phrase over and over again in a silly, sing-song voice. The repetition of the phrase is humorous, and the speaker's enthusiasm for wanting to go to the aquarium is endearing. The way they keep asking "somebody" to take them to the aquarium is especially funny. |
| **MAF** | It's funny because the man is talking about going to the aquarium, but he says he wants to go to someone else's aquarium. |
| **T5 w/ P** | A man is holding a goldfish in his hand and he says "Aquarium, aquarium, I wanna go to the aquarium." The fish then starts to squirm and he says "Okay, aquarium, somebody." It is funny because the man is trying to get the fish to go to the aquarium, but the fish is not interested in going. |
| **BART w/ P** | It's funny because the narrator keeps saying "aquarium" over and over again, but the fish doesn't seem to care. |
| **GPT-3.5 w/ P** | The video is funny because the goldfish is portrayed as a character that wants to go to the aquarium. The goldfish is seen swimming around the tank and looking up at the camera with its big eyes, as if it is asking for help. The video ends with an ad for the Oregon Coast Aquarium in Newport, making it even funnier. |
| **Gold** | The goldfish is swimming around in his bowl saying, "I want to go to the aquarium." It is funny because he is a goldfish and he's already in a tank full of water. |

🎙️ I like the color. You like the color? Yeah. I had a mixed reviews on the color. Some people said they like it, some people said they don't like it. And I said, well what color is your baguette?

| | |
|---|---|
| **T5** | The man is describing a baguette that he bought and he says he liked the color, and then he says "some people said they like it, some people said they don't like it." It is funny because he is implying that people don't like the color of the baguette. |
| **BART** | A man asks a woman what color is her baguette. She replies, "I had a mixed reviews on the color, some people said they don't like it." The man then asks her, "And I said, well what color?" It is funny because it is unexpected. |
| **GPT-3.5** | The speaker's response to the question of whether people like the color or not is hilarious. The unexpected and absurd question of what color their baguette is catches the audience off guard and elicits a burst of laughter. The speaker's deadpan delivery of the line only adds to the humor. |
| **MAF** | A man says "I had a mixed reviews on the color of my baguette." It is funny because it is unexpected. |
| **T5 w/ P** | The man in the cartoon says "I like the color, you like the color?" and then the car is towed away. It's funny because the car is a very bright color and the man in the cartoon is a very nerdy looking guy. |
| **BART w/ P** | A man says "I had a mixed reviews on the color" while looking at a car. This is funny because it's a play on the word mixed. |
| **GPT-3.5 w/ P** | The video is funny because the speaker is making a joke about the color of the car, asking what color the person's baguette is in comparison. The visuals add to the humor, showing the man in sunglasses and orange shirt next to the orange sports car, and the tow truck driver with his tool. The sound of the car at the beginning also adds to the comedic effect. |
| **Gold** | The video is funny because it features an animated clip featuring a man that is towing a luxury car while in the presence of its presumably wealthy owner that makes the sly comment in stating, "I had mixed reviews on the color. Some people say they like it, some people said they don't like it. I said well, what color is your Baguette?" This remark is laughable because it implies that the man is rich and the person that is questioning the car color is not and is likely not in ownership of a Baguette, so therefore, their opinion is irrelevant. |

Figure 15: **(Correct)** An example of explanation generation results. Except for LLMs with our prompting, the presence of a goldfish (or fish) is not mentioned. Particularly, GPT-3.5 with our prompting accurately describes the character and behavior of the goldfish, and even mentions the content of the concluding advertisement.

Figure 16: **(Correct)** An example of explanation generation results. Text-only LLMs do not mention a car that has a similar color to a Baguette. Meanwhile, LLMs with our prompting provide details about the car in the scene. Note that GPT-3.5 with our prompting can explain the sarcasm related to the baguette like in Gold.

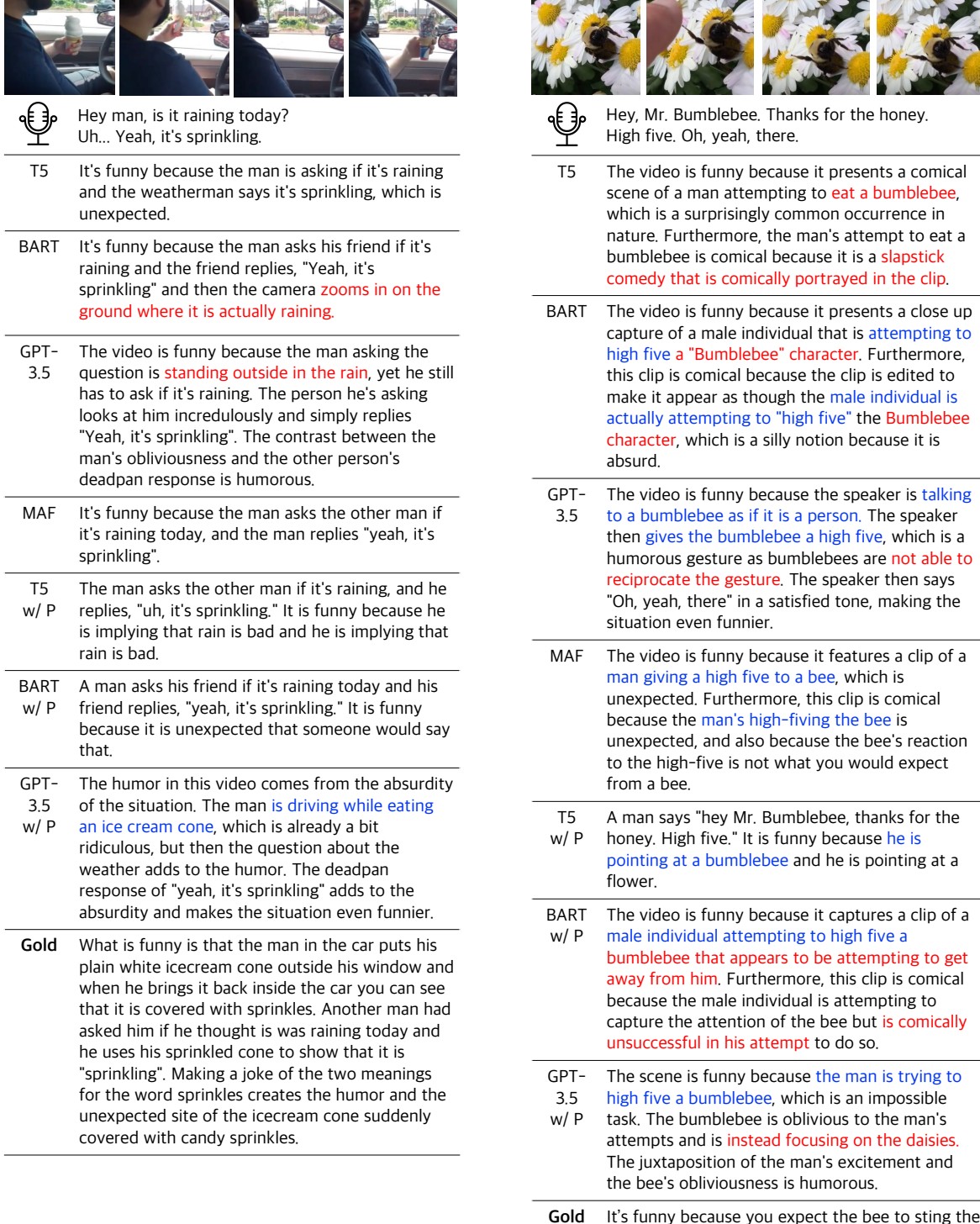

🎙️ Hey man, is it raining today?
Uh... Yeah, it's sprinkling.

| | |
|---|---|
| T5 | It's funny because the man is asking if it's raining and the weatherman says it's sprinkling, which is unexpected. |
| BART | It's funny because the man asks his friend if it's raining and the friend replies, "Yeah, it's sprinkling" and then the camera zooms in on the ground where it is actually raining. |
| GPT-3.5 | The video is funny because the man asking the question is standing outside in the rain, yet he still has to ask if it's raining. The person he's asking looks at him incredulously and simply replies "Yeah, it's sprinkling". The contrast between the man's obliviousness and the other person's deadpan response is humorous. |
| MAF | It's funny because the man asks the other man if it's raining today, and the man replies "yeah, it's sprinkling". |
| T5 w/ P | The man asks the other man if it's raining, and he replies, "uh, it's sprinkling." It is funny because he is implying that rain is bad and he is implying that rain is bad. |
| BART w/ P | A man asks his friend if it's raining today and his friend replies, "yeah, it's sprinkling." It is funny because it is unexpected that someone would say that. |
| GPT-3.5 w/ P | The humor in this video comes from the absurdity of the situation. The man is driving while eating an ice cream cone, which is already a bit ridiculous, but then the question about the weather adds to the humor. The deadpan response of "yeah, it's sprinkling" adds to the absurdity and makes the situation even funnier. |
| **Gold** | What is funny is that the man in the car puts his plain white icecream cone outside his window and when he brings it back inside the car you can see that it is covered with sprinkles. Another man had asked him if he thought is was raining today and he uses his sprinkled cone to show that it is "sprinkling". Making a joke of the two meanings for the word sprinkles creates the humor and the unexpected site of the icecream cone suddenly covered with candy sprinkles. |

🎙️ Hey, Mr. Bumblebee. Thanks for the honey.
High five. Oh, yeah, there.

| | |
|---|---|
| T5 | The video is funny because it presents a comical scene of a man attempting to eat a bumblebee, which is a surprisingly common occurrence in nature. Furthermore, the man's attempt to eat a bumblebee is comical because it is a slapstick comedy that is comically portrayed in the clip. |
| BART | The video is funny because it presents a close up capture of a male individual that is attempting to high five a "Bumblebee" character. Furthermore, this clip is comical because the clip is edited to make it appear as though the male individual is actually attempting to "high five" the Bumblebee character, which is a silly notion because it is absurd. |
| GPT-3.5 | The video is funny because the speaker is talking to a bumblebee as if it is a person. The speaker then gives the bumblebee a high five, which is a humorous gesture as bumblebees are not able to reciprocate the gesture. The speaker then says "Oh, yeah, there" in a satisfied tone, making the situation even funnier. |
| MAF | The video is funny because it features a clip of a man giving a high five to a bee, which is unexpected. Furthermore, this clip is comical because the man's high-fiving the bee is unexpected, and also because the bee's reaction to the high-five is not what you would expect from a bee. |
| T5 w/ P | A man says "hey Mr. Bumblebee, thanks for the honey. High five." It is funny because he is pointing at a bumblebee and he is pointing at a flower. |
| BART w/ P | The video is funny because it captures a clip of a male individual attempting to high five a bumblebee that appears to be attempting to get away from him. Furthermore, this clip is comical because the male individual is attempting to capture the attention of the bee but is comically unsuccessful in his attempt to do so. |
| GPT-3.5 w/ P | The scene is funny because the man is trying to high five a bumblebee, which is an impossible task. The bumblebee is oblivious to the man's attempts and is instead focusing on the daisies. The juxtaposition of the man's excitement and the bee's obliviousness is humorous. |
| **Gold** | It's funny because you expect the bee to sting the man but the bee gives the man a high five instead and then puts his arm back. |

Figure 17: **(Incorrect)** An example of explanation generation results. Unlike both text-only BART and GPT-3.5 say it's raining, GPT-3.5 with our prompting correctly mentions a man eating an ice cream cone in his car. However, it fails to explain the use of the "sprinkled cone" for a pun.

Figure 18: **(Incorrect)** An example of explanation generation results. Unlike text-only LLMs, both MAF and LLMs with our prompting correctly identify Bumblebee as a bee, not a character. However, they incorrectly generate explanations saying that Bumblebee fails to perform a "high-five," which differs from Gold.