# OpenReview forum: "Can Language Models Laugh at YouTube Short-form Videos?"
_EMNLP/2023/Conference — EMNLP 2023 Main_

### Official Review · Reviewer_gSbC · 2023-08-01

**Soundness:** 5

**Excitement:**

4: Strong: This paper deepens the understanding of some phenomenon or lowers the barriers to an existing research direction.

**Paper Topic And Main Contributions:**

This paper utilizes LLMs to understand the humorous content within videos by harnessing multimodal features. The main contributions of this study are as follows:

1. The authors have curated the ExFunTube dataset, where both textual and visual features play a crucial role in conveying humor. This compilation was achieved through an innovative data selection pipeline.
2. The authors have developed a zero-shot video-to-text prompting framework specifically designed to optimize the LLM's comprehension of humor within videos.

**Questions For The Authors:**

A. Were there instances where videos were identified by the MTurk annotators as non-humorous but weren't eliminated by the pipeline? This could provide a quantitative assessment of the pipeline's efficiency.

B. Can you specify the number of annotators per video? Was there only one annotator for each video, or were multiple involved?

C. How adaptable do you believe your framework is to other datasets of different topics? Could you shed some light on potential difficulties or nuances that might arise during such generalization?

**Reasons To Accept:**

This paper has many strengths that make it worthy of acceptance, including the following:

1. The authors have devised an innovative and inspiring method for dataset creation, seamlessly capturing both textual and visual nuances.
2. By leveraging current state-of-the-art (SOTA) models and techniques, the paper showcases its relevance and modern approach.
3. The team has conducted thorough experiments and evaluations.
4. The research's modular framework stands out, designed to easily integrate or replace components with upcoming SOTA models, ensuring long-term adaptability.

**Reasons To Reject:**

The paper exhibits promising strengths; however, there are certain aspects that could be improved:

1. Audio information is not fully utilized. A more extensive exploitation of this modality could enhance the system's comprehension of the data and potentially improve the overall results.
2. The choice of selected models and methods in the study could use more detailed justification. Understanding why these specific models were chosen over others and their relative advantages would be beneficial. Additionally, the potential pitfalls or errors introduced by these models are not sufficiently addressed.

**Reproducibility:**

4: Could mostly reproduce the results, but there may be some variation because of sample variance or minor variations in their interpretation of the protocol or method.

**Reviewer Confidence:**

4: Quite sure. I tried to check the important points carefully. It's unlikely, though conceivable, that I missed something that should affect my ratings.

---

> ### Author Rebuttal · Authors · 2023-08-29
>
> We thank the reviewer for all feedback and positive comments. We address all concerns below.
>
> **Reasons to reject 1. Underutilization of audio information.**
>
> We divided the audio information into speech and sound components. From the speech component, we extracted transcripts, and from the sound component, we obtained sound tags. We temporally aligned the speech with the visual component using timestamps, but we did not with the sound tags, which could be seen as not fully utilizing the audio. However, it was based on our experiments that putting sound tags for the whole video led to better performance. We tested temporally segmenting the whole audio input and adding sound tags per segment, but it was unstable for training and led to information loss and hallucinations.
>
> **Reasons to reject 2. Lack of detailed justification for model selection and insufficient discussion on potential pitfalls.**
>
> Our prompting method aims at extracting as much information as possible from YouTube videos (diverse user-generated videos) into text in a zero-shot manner. Therefore, we apply SOTA zero-shot models for each modality. For example, in the dataset collection pipeline, we used a Zero-Shot Video Captioning model [1] for video captioning, which adopted GPT-2 [2] and CLIP [3] that were pretrained on large-scale datasets. In prompting, for speech understanding, we labeled the speaker using ChatGPT with the transcript obtained from Whisper. We also tried other models, for example, mPLUG-2 [4] and Swinbert [5] for zero-shot video captioning, OFA [6] and mPLUG [7] for image captioning, GRiT [8] and GLIPv2 [9] for object detection, CLIP4Clip [10] for video-to-text retrieval, Pyannote [11] for speaker diarization, and CAV-MAE (audio-only) [12] for sound tagging, but they suffered from more information loss and hallucination than the models we chose.
>
> Admittedly, there were other issues when we used our chosen models. The Zero-Shot Video Captioning model took 10 minutes per video because it repeated the process of CLIP loss guidance to GPT-2 at inference time. In total, it took one week with 8 A6000 GPUs to caption all candidate videos before passing through the pipeline. Also, BLIP-2 took 48 hours with 8 A6000 GPUs to extract captions from video frames. For speaker diarization, ChatGPT sometimes made errors because it is given only the transcript, but it worked better than other models. Finally, unlike other models, EfficientAT was not robust to our dataset because it was not pretrained on sufficiently large-scale datasets.
>
> In the final draft, we will add the detailed rationale for choosing each model and the limitations that could occur.
>
>
> **Question 1. Non-humorous instances for assessment of pipeline’s efficiency.**
>
> During the dataset collection, MTurk workers also identified the videos that had no funny moments. The number of such videos is 453 (about 4%), which indicates that our pipeline is effective.
>
>
> **Question 2. Number of annotators per video.**
>
> Each video was annotated by a single worker. We did not obtain multiple annotations per video, because it would take a lot of time and cost for large-scale video sets. Instead, we carefully selected annotators who could provide high-quality explanations, and we verified every annotation by ourselves. As stated in Appendix C, MTurk workers must meet high standards to participate in our dataset annotation: a HIT approval rate of 99% or higher, a total of more than 10,000 approved HITs, and be located in one of the countries of AU, CA, GB, NZ, or US. In our qualification test, we only selected the workers who could explain humor well in funny videos. Only 60 workers out of 219 passed the qualification test, indicating our rigorous and careful selection. More crowdsourcing details can be found in Appendix C.
>
>
> **Question 3. Adaptability and potential challenges of the framework in generalization.**
>
> Our prompting framework, designed to extract as much information from videos into text as possible, can be applied to multimodal datasets of different topics, including three types of TV shows, sitcoms, medical dramas, and crime dramas, in TVQA [13], and short clips from movies in LSMDC [14].
>
> However, it may have some difficulties in certain types of videos. To extract visual information, we divided the video into multiple segments using utterance timestamps and scene boundary detection. Therefore, videos with little utterances and infrequent scene changes (e.g., Kinetics [15]) may not be easily segmented by our method. Additionally, we obtained sound tags from the entire audio stream for considering non-verbal auditory effects. This may not be suitable for datasets that require detailed information about sound (e.g., MUSIC-AVQA [16]).
>
>
> **References**
>
> [1] Tewel, Yoad, et al. "Zero-shot video captioning with evolving pseudo-tokens." arXiv preprint arXiv:2207.11100 (2022).
> [2] Radford, Alec, et al. "Language models are unsupervised multitask learners." OpenAI blog 1.8 (2019): 9.
> [3] Radford, Alec, et al. "Learning transferable visual models from natural language supervision." International conference on machine learning. PMLR, 2021.
> [4] Xu, Haiyang, et al. "mPLUG-2: A modularized multi-modal foundation model across text, image and video." arXiv preprint arXiv:2302.00402 (2023).
> [5] Lin, Kevin, et al. "Swinbert: End-to-end transformers with sparse attention for video captioning." Proceedings of the IEEE/CVF Conference on Computer Vision and Pattern Recognition. 2022.
> [6] Wang, Peng, et al. "Ofa: Unifying architectures, tasks, and modalities through a simple sequence-to-sequence learning framework." International Conference on Machine Learning. PMLR, 2022.
> [7] Li, Chenliang, et al. "mPLUG: Effective and Efficient Vision-Language Learning by Cross-modal Skip-connections." Proceedings of the 2022 Conference on Empirical Methods in Natural Language Processing. 2022.
> [8] Nguyen, Van-Quang, Masanori Suganuma, and Takayuki Okatani. "Grit: Faster and better image captioning transformer using dual visual features." European Conference on Computer Vision. Cham: Springer Nature Switzerland, 2022.
> [9] Zhang, Haotian, et al. "Glipv2: Unifying localization and vision-language understanding." Advances in Neural Information Processing Systems 35 (2022): 36067-36080.
> [10] Luo, Huaishao, et al. "Clip4clip: An empirical study of clip for end to end video clip retrieval and captioning." Neurocomputing 508 (2022): 293-304.
> [11] Bredin, Hervé, et al. "Pyannote. audio: neural building blocks for speaker diarization." ICASSP 2020-2020 IEEE International Conference on Acoustics, Speech and Signal Processing (ICASSP). IEEE, 2020.
> [12] Gong, Yuan, et al. "Contrastive audio-visual masked autoencoder." The Eleventh International Conference on Learning Representations. 2022.
> [13] Lei, Jie, et al. "TVQA: Localized, Compositional Video Question Answering." Proceedings of the 2018 Conference on Empirical Methods in Natural Language Processing. 2018.
> [14] Rohrbach, Anna, et al. "Movie description." International Journal of Computer Vision 123 (2017): 94-120.
> [15] Carreira, Joao, and Andrew Zisserman. "Quo vadis, action recognition? a new model and the kinetics dataset." proceedings of the IEEE Conference on Computer Vision and Pattern Recognition. 2017.
> [16] Li, Guangyao, et al. "Learning to answer questions in dynamic audio-visual scenarios." Proceedings of the IEEE/CVF Conference on Computer Vision and Pattern Recognition. 2022.

---

### Official Review · Reviewer_k1U2 · 2023-08-02

**Typos Grammar Style And Presentation Improvements:** N/A
**Soundness:** 3

**Ethical Concerns:**

Yes

**Excitement:**

4: Strong: This paper deepens the understanding of some phenomenon or lowers the barriers to an existing research direction.

**Justification For Ethical Concerns:**

There is some uncomfortable content in the dataset.

**Missing References:**

N/A

**Paper Topic And Main Contributions:**

The article released a multimodal humor video dataset. It uses mature tools to convert visual and audio signals into text signals, and is manually annotated. The videos in this dataset contain one or more humor points, and experiments have shown that although existing large models can effectively identify humor points, there is still room for improvement.

**Questions For The Authors:**

N/A

**Reasons To Accept:**

1. The article released a large-scale multimodal humor video dataset that provides novel multi-humor interval annotations, which can provide new data support for community understanding of videos.
2. Although there are numerous similar frameworks, the article provides a simple and feasible method for converting videos to text. It can capture videos according to their plot/shot.
3. The article provides a feasible method for understanding humorous videos, which can achieve the SOTA understanding performance.

**Reasons To Reject:**

1. The article provides humor interval annotations, but the distribution of the number of humor points is not very balanced, and I believe that videos without humor intervals should be added (negative samples can assist the model in understanding the true meaning of humor).
2. The annotation of the released dataset was not fully evaluated for quality (or not displayed in the article), so the quality of the dataset cannot be guaranteed.
3. There is uncomfortable content in the released dataset. For example, the second example in the attachment not only has formatting issues, but its content also makes me feel very uncomfortable while watching.

**Reproducibility:**

4: Could mostly reproduce the results, but there may be some variation because of sample variance or minor variations in their interpretation of the protocol or method.

**Reviewer Confidence:**

4: Quite sure. I tried to check the important points carefully. It's unlikely, though conceivable, that I missed something that should affect my ratings.

---

> ### Author Rebuttal · Authors · 2023-08-29
>
> We thank the reviewer for the helpful comments and for addressing all concerns below.
>
> **Reasons to reject 1. Unbalanced distribution of the number of humor points and need for negative samples.**
>
> In our dataset, the distribution of humor intervals over videos is not balanced, but we believe this is not a big issue. Each video includes varied content, so it can have any number of humor intervals. Furthermore, the videos in our dataset are short-form, often less than 30 seconds, it is more common for each video to have one humor interval. The purpose of collecting multiple intervals per video is to ensure that annotators capture all the humorous moments and document all of them to avoid missing.
>
> We thank the reviewer for the suggestion about negative samples. During the dataset collection, MTurk workers also identified the videos that had no funny moments. The number of such videos is 453 (about 4%), and they can be used as negative samples.
>
> Or, one can use the parts of videos that are not included in the ‘funny moments’ as negative samples. We will update our manuscript to discuss this negative sample issue clearer.
>
>
>
> **Reasons to reject 2. Concerns over unguaranteed dataset quality.**
>
> As stated in Appendix C, MTurk workers must meet high standards to participate in our dataset annotation: a HIT approval rate of 99% or higher, a total of more than 10,000 approved HITs, and be located in one of the countries of AU, CA, GB, NZ, or US. In our qualification test, we only selected the workers who could explain humor well in funny videos. Only 60 workers out of 219 passed the qualification test, indicating our rigorous and careful selection.
>
> It is challenging to guarantee the quality of free-form annotations. Therefore, we let the workers first annotate timestamps to identify what content is funny, and then write a detailed answer of at least 100 characters about the moment they found funny. Furthermore, as mentioned in Section 3.2, we asked each annotator to write in the format of “[What is funny]. It is funny because [Explanation of why funny]” to ensure they not only describe the funny moment but also elaborate on the reason why it is funny. We removed the responses that do not follow this direction. Workers who continued to disregard this standard were excluded from further annotation.
> Responses that include both the description and the reason were individually reviewed by the authors to verify whether the reason for humor was valid, and if not agreed upon, the responses were collected again. The human evaluation score of the annotated explanation was 0.792, as shown in Table 2, which fell between weak yes (0.75) and yes (1), demonstrating the appropriateness of the annotation.
>
> Furthermore, when we performed human evaluation, we asked workers to check all shortcomings of the explanation. The shortcomings include Missed Humor, Too Vague, Incorrect Details, Lacking Context, and Too Verbose; please see Figure 7 in Appendix for descriptions of these shortcomings. For randomly sampled 100 videos, we asked five annotators per video to select all shortcomings that existed and select None if there was no shortcoming. The presence of each shortcoming was determined by a majority vote.
>
> The results are shown below. For Gold explanations, instances of Missed Humor or Incorrect Details are much less than model-generated explanations. Also, instances of Too Vague and Lacking Context are less than model-generated explanations except GPT-3.5 w/ Prompting. It seems that GPT-3.5 w/ Prompting includes much information to explain humor, which causes the trade-off between vagueness/lacking context and verbosity (19 instances of Too Verbose for GPT-3.5 w/ Prompting). Lastly, there are significantly more instances of None for Gold explanations than for model-generated explanations and about three times more than GPT-3.5 w/ Prompting. This suggests that Gold explanations show high-level quality.
>
> ||Missed Humor|Too Vague|Incorrect details|Lacking Context|Too Verbose|None|
> |---|:---:|:---:|:---:|:---:|:---:|:--:|
> |Text-only BART|75|17|69|27|24|0|
> |Text-only GPT-3.5 |56|25|40|33|2|4|
> |MAF|76|21|72|47|4|0|
> |BART w/ Prompting|62|18|64|34|7|1|
> |GPT-3.5 w/ Prompting|29|7|48|15|19|11|
> |Gold|10|15|3|20|2|38|
>
> We additionally conduct human evaluations for randomly sampled 100 videos to verify the quality of annotation. Based on the criteria of good explanations that [1] suggested, which are Circularity, Relevance, and Coherence, we ask three annotators per video to respond with either Yes or No. The results are shown below. High Circularity, Relevance, and Coherence scores show considerable quality of annotations. Since the rebuttal period is short, we evaluate only 100 examples, but we will extend it to the whole dataset in the final version.
>
> ||Circularity|Relevance|Coherence|
> |---|:---:|:---:|:---:|
> |Score|0.94|1.0|0.98|
>
>
>
> **Reasons to reject 3. Uncomfortable content in the dataset.**
>
> We manually reviewed videos based on the safety objectives proposed by [2] from Google Research, as mentioned in the Ethics statement. We rigorously filtered out any videos with ethical issues; as a result, we chose only 10K videos out of the 21K ones that passed through our pipeline (50% were filtered). The second example that the reviewer felt uncomfortable has no ethical issue according to our filtering criteria. In the video, three characters emotionlessly repeated “That’s so funny” in irregular timings. However, we totally understand that it might be uncomfortable for someone since the characters may look somewhat bizarre. We will recheck all the videos again and see if there are any videos that might cause some discomfort.
>
> **References**
>
> [1] Von Wright, Georg Henrik. Explanation and understanding. Cornell University Press, 2004.
> [2] Thoppilan, Romal, et al. "Lamda: Language models for dialog applications." arXiv preprint arXiv:2201.08239 (2022).

---

### Official Review · Reviewer_3agr · 2023-08-03

**Typos Grammar Style And Presentation Improvements:** “curate” doesn't seem like a common w…
**Soundness:** 3

**Excitement:**

2: Mediocre: This paper makes marginal contributions (vs non-contemporaneous work), so I would rather not see it in the conference.

**Paper Topic And Main Contributions:**

For processing multi-modal video humor explanation, this paper constructs a multi-modal humor video dataset ExFunTube from YouTube with high quality control and support from large language model (LLM). This dataset ExFunTube cover a wide range of domains with various types of humor, and contains annotate start and end timestamps of funny moments and provide text explanations for each moment. Then, this paper proposes a zero-shot prompt approach to explain video humor via LLM’s Excellent semantic understanding ability. Three different evaluation methods seem to demonstrate the approach improves LLM's performance on humor explanation.

**Questions For The Authors:**

1.	Although binary classifications or ratings do not fully evaluate whether a model understands humor in detail, have you made any relevant attempts? And whether the LLM-labelled humor samples exhibit LLM-specific preferences?
2.	You believe that a flaw of the dataset in MAF is the reliance on visual information (at line 79). Why do you spend much resource to mine visual modality with BLIP-2?
3.	What are the prompt demonstrations you used in the experiments?
4.	What is the difference between speech and sound contexts? Which is more important?
5.	Are there other prompt-based baselines for textual description generation?
6.	What is the time complexity? Where is the implementation details?

**Reasons To Accept:**

1.	Construct a multi-modal humor video dataset ExFunTube from YouTube with high quality control and support from large language model(LLM), the dataset cover a wide range of domains with various types of humor, and contains annotate start and end timestamps of funny moments and provide text explanations for each moment. This seems the only contribution.

**Reasons To Reject:**

1.	The author believes that a flaw of the dataset in MAF is the reliance on visual information (at line 79). However, my opinion is exactly the opposite. Since it is a multimodal data set, it is reasonable to rely on any modality, and each modality has its importance.
2.	Due to the conversion of visual and acoustic modalities into text descriptions, the final explanation of humor is entirely dependent on text modality, which leads to the loss of some important modal-specific information in the intermediate steps. More appropriate feature encoding and modal fusion methods could be improved.
3.	In section 4.1, I didn’t see any prompt demonstration. And I didn’t understand why you emphasize the performance improvement (Line 334) in the section 4.1, which is the part of approach, rather than experimentation.
4.	The proposed approach relies on tedious preprocessing steps for three modalities. For example, BLIP-2 needs much computing resource and time. And simultaneously with GPT3.5, the application of this approach is very unrealistic.
5.	Some comparisons with various prompt baselines (same LLM) are missing.
6.	Missing the experiments and analysis about the importance of visual, speech and sound in your task.
7.	With LLMs, it seems natural for your proposed approach performing better than MAF with only traditional Transformer.
8.	This paper is hard to follow and the experiments are difficult to re-produce.

**Reproducibility:**

2: Would be hard pressed to reproduce the results. The contribution depends on data that are simply not available outside the author's institution or consortium; not enough details are provided.

**Reviewer Confidence:**

4: Quite sure. I tried to check the important points carefully. It's unlikely, though conceivable, that I missed something that should affect my ratings.

---

> ### Author Rebuttal · Authors · 2023-08-29
>
> We thank the reviewer for the detailed feedback and address all comments below.
>
>
> **Reasons to reject 1. & Question 2. Disagreement on the reliance on visual information in the dataset of MAF.**
>
> We feel very unfortunate that there is a big misunderstanding about our description of the MAF dataset. In line 79, we mentioned that “However, this dataset is limited to sarcasm, a specific form of humor, and focuses on sarcasm explanation in dialogue. It highlights a need for a humor explanation dataset that considers visual elements and covers general humor.” We never said that there is a flaw in the MAF, but pointed out that it only dealt with sarcasm as a specific form of humor, and thus a dataset that covers general humor is needed like our ExFunTube dataset. Also, we never said that it relies on visual information, but simply said it focuses more on dialogue. In fact, the MAF paper named its target task as the “Sarcasm Explanation in Dialogue” task.
>
>
> **Reasons to reject 2. Loss of modal-specific information due to conversion into text description.**
> Our paper title is “Can LLMs understand video humor?” Since most powerful LLMs, including GPT-3.5 and 4, can receive only text input, we have no choice but to convert the video content into text. GPT-4 is supposed to be multimodal by enabling an image as input; as of the paper submission, it could not accept images as input. Even if we use feature encoding and model fusion methods, in the end, we have to convert it into text to exploit LLM’s incredible ability of explanation generation. In both cases, information loss is inevitable when we have to provide the video content as text input to LLMs. Therefore, we explored a prompting method that can minimize such information loss.
>
>
> **Reasons to reject 3. & Question 3. No prompt demonstration and misplaced emphasis on performance improvement.**
> The prompt demonstration is shown in Figure 3. (b). The reason for mentioning performance improvement in line 334 is to hint that the effectiveness of our prompting would be validated in the experiments in Section 5.
>
>
> **Reasons to reject 4. & Question 6. Concerns about the resource intensity of the proposed approach.**
> This work focuses on evaluating whether LLMs can understand the humor in the videos. Given that we were going to use existing LLMs, our methodological focus was how to generate and configure a text prompt that can maximally extract LLM’s capability for humor explanation. Since the multimodal information in a video should be converted into text as an input to LLMs with minimal information loss, our choice was to use state-of-the-art (SOTA) models, even if it requires some amount of computing resources. In the case of BLIP-2, we extracted time (sec) * 5 frames/sec * 20 captions/frame captions for each video, so video captioning took about 48 hours for our whole dataset using 8 A6000 GPUs. Also, speaker diarization with GPT-3.5-turbo took 48 hours, and explanation generation with GPT-3.5 text-davinci-003 took 24 hours. Considering the size of the dataset, this amount of computing resources and time is not unrealistic. Please see Appendix A for more implementation details asked in Question 6.
>
>
> **Reasons to reject 5. & Question 5. Lack of comparisons with various prompt baselines.**
> The results of various prompt baselines using the same LLM are presented in Table 2 and Table 3. The Text-only in Table 2 indicates the prompt baseline using only transcripts obtained from speech. Additionally, Table 3 shows the results of our approach excluding one component each from visual, speech, and sound.
>
> During the rebuttal period, we conduct additional experiments with another multimodal prompt baseline VideoChat* [1]. It extracts information from videos using two modules: (i) VideoChat-Text generates text descriptions from the video, and (ii) VideoChat-Embed encodes the video as embeddings. The text descriptions and embeddings are combined with the received question as a prompt given to the LLM to generate a response. VideoChat uses the same  LLM (GPT-3.5 text-davinci-003) as our method, and the question given to the LLM is “Explain why the video is funny in up to 3 sentences.” The results can be seen in the table below. Our prompting method scores significantly higher than VideoChat by 0.150 in SentBERT Mean and 0.069 in ROSCOE (RA) Mean.
>
> *The VideoChat-Text module adopts Tag2Text [2], GRiT [3], and T5, in addition to InternVideo and Whisper that are also used in our prompting. Moreover, the VideoChat-Embed module includes StableVicuna [4] and the temporal modeling of InternVideo, in addition to BLIP-2 used in our prompting. That is, the VideoChat baseline is much heavier than our prompting with worse performance.
>
>
>
> ||SentBERT @0.7|SentBERT @0.6|SentBERT @0.5|SentBERT @0.4|SentBERT Mean|ROSCOE (RA) @0.8|ROSCOE (RA) @0.7|ROSCOE (RA) Mean|
> |---|---|---|---|---|---|---|---|---|
> |Videochat w/ GPT-3.5|0.020|0.115|0.344|0.671|0.452|0.197|0.783|0.748|
> |MAF|0.149|0.375|0.604|0.809|0.541|0.438|0.880|0.785|
> |Our-prompting  w/ GPT-3.5 |0.214|0.541|0.806|0.945|0.602|0.639|0.971|0.817|
>
>
> **Reasons to reject 6. Missing ablation studies.**
> We conducted ablation experiments; please see Table 3 in the main draft for the results of GPT-3.5 with removing each of the visual, speech, and sound components. We also discussed these results in Section 5.6. Moreover, the results of using T5 and BART as the backbone LLM were presented in Table 4 in Appendix and discussed in Appendix B.
>
>
> **Reasons to reject 7. LLM with prompting naturally surpasses Transformer.**
> The table above shows that the VideoChat prompting with GPT-3.5 underperformed MAF in terms of SentBERT and ROSCOE (RA) scores. It implies that simply adopting a strong LLM does not guarantee higher performance, and our prompting method accurately converts the video content into text and makes GPT-3.5 better explain the humor in the videos.
> Furthermore, Table 2 already showed that the BART with our prompting performed better than MAF. Please note that MAF also used BART-Large as a backbone with additional modules. In Table 2, BART with our prompting had a higher SentBERT Mean score by 0.047, ROSCOE (RA) Mean by 0.02, Rationale Quality score by 1.4, and Human Evaluation Rating by 0.151.
>
>
> **Reasons to reject 8. Low reproducibility.**
> The implementation details are provided in Appendix A, and the proposed prompting does not require additional training. We will also release the code for dataset collection and our prompting method.
>
>
>
> **Question 1. Attempt at binary classification and presence of LLM-specific preferences.**
> Since our target task is humor explanation in natural language, we did not try binary classifications or ratings. Recent studies ([5], [6]) also argued that moving forward from identification is necessary. Our dataset is fully labeled by humans; the funny videos were verified by AMT workers, and explanation sentences were also written by human workers.
>
>
> **Question 4. Difference between speech and sound components.**
> Speech refers to utterances, while sound refers to other non-verbal auditory effects. For instance, in the example of Figure 1, the utterances represented by the blue bubble are speech, while the sound is the barking of a dog or the sound of panting. Figure 3. (b) shows how speech and sound are used in our prompt, where speech includes utterances like “Hey Luke, sit. Luke, dandelion. AHHHH!”, and sound includes sound tags like “Speech, Panting, Dog”. The ablation results in Table 3 show that speech is more important than sound, as the w/o T baseline (excluding speech) led to a more significant performance drop than the w/o A baseline (excluding sound).
>
>
>
> **References**
>
> [1] Li, KunChang, et al. "Videochat: Chat-centric video understanding." arXiv preprint arXiv:2305.06355 (2023).
> [2] Huang, Xinyu, et al. "Tag2text: Guiding vision-language model via image tagging." arXiv preprint arXiv:2303.05657 (2023).
> [3] Nguyen, Van-Quang, Masanori Suganuma, and Takayuki Okatani. "Grit: Faster and better image captioning transformer using dual visual features." European Conference on Computer Vision. Cham: Springer Nature Switzerland, 2022.
> [4] StableLM contributors. Stablelm: Stability ai language models. stableLM, 2023.
> [5] Kumar, Shivani, et al. "When did you become so smart, oh wise one?! Sarcasm Explanation in Multi-modal Multi-party Dialogues." Proceedings of the 60th Annual Meeting of the Association for Computational Linguistics (Volume 1: Long Papers). 2022.
> [6] Chakrabarty, Tuhin, et al. "Flute: Figurative language understanding through textual explanations." Proceedings of the 2022 Conference on Empirical Methods in Natural Language Processing. 2022.

---

### Meta-Review · Area_Chair_FTSA · 2023-09-18

**Recommendation:** 4

**Metareview:**

All reviewers appreciated the dataset introduced by this paper, finding it "wide-domain", "large-scale", and created through an "innovative data selection pipeline". Reviewers generally agreed that the work is sound, with some concerns about possible data quality issues from reviewer k1U2 addressed in the author response. Reviewer 3agr gave a much lower excitement score than other reviewers, however I tend to agree with the other reviewers that the combination of model generations and human selection, and evaluation, will likely make this a high quality resource, and that the experiments done here look very thorough.

To address the points related to dataset discomfort raised by reviewer k1U2, I strongly recommend that the ​​authors ensure that details of their manual review process and use of the Lamda guidelines are clear in the paper, and also recommend (but less strongly) that they write a content warning for readers of the paper, and users of the dataset, about possible discomfort caused by some of the videos in the dataset.

---

### Decision · Program_Chairs · 2023-10-07

**Decision:**

Accept-Main

**Comment:**

All reviewers appreciated the dataset introduced by this paper, finding it "wide-domain", "large-scale", and created through an "innovative data selection pipeline". Reviewers generally agreed that the work is sound, with some concerns about possible data quality issues from reviewer k1U2 addressed in the author response. Reviewer 3agr gave a much lower excitement score than other reviewers, however I tend to agree with the other reviewers that the combination of model generations and human selection, and evaluation, will likely make this a high quality resource, and that the experiments done here look very thorough.

To address the points related to dataset discomfort raised by reviewer k1U2, I strongly recommend that the ​​authors ensure that details of their manual review process and use of the Lamda guidelines are clear in the paper, and also recommend (but less strongly) that they write a content warning for readers of the paper, and users of the dataset, about possible discomfort caused by some of the videos in the dataset.